# Attention operates uniformly throughout the classical receptive field and the surround

**Bram-Ernst Verhoef[1,2,3]\*, John HR Maunsell[1,2]**

[1]Department of Neurobiology, The University of Chicago, Chicago, United States;
[2]Department of Neurobiology, Harvard Medical School, Boston, United States;
[3]Laboratorium voor Neuro- en Psychofysiologie, Katholieke Universiteit Leuven, Leuven, Belgium

**Abstract** Shifting attention among visual stimuli at different locations modulates neuronal responses in heterogeneous ways, depending on where those stimuli lie within the receptive fields of neurons. Yet how attention interacts with the receptive-field structure of cortical neurons remains unclear. We measured neuronal responses in area V4 while monkeys shifted their attention among stimuli placed in different locations within and around neuronal receptive fields. We found that attention interacts uniformly with the spatially-varying excitation and suppression associated with the receptive field. This interaction explained the large variability in attention modulation across neurons, and a non-additive relationship among stimulus selectivity, stimulus-induced suppression and attention modulation that has not been previously described. A spatially-tuned normalization model precisely accounted for all observed attention modulations and for the spatial summation properties of neurons. These results provide a unified account of spatial summation and attention-related modulation across both the classical receptive field and the surround.

**\*For correspondence:** verhoef@uchicago.edu

**Competing interests:** The authors declare that no competing interests exist.

## Introduction

Our eyes are constantly bombarded by a welter of visual stimuli, only a small fraction of which can be processed thoroughly (*Chun et al., 2011*; *Kastner and Ungerleider, 2000*). Spatial attention sifts through the plethora of stimuli – enhancing perception at behaviorally-relevant locations – but the underlying neural principles of this process are not fully understood (*Chun et al., 2011*; *Kastner and Ungerleider, 2000*; *Posner, 1980*; *Carrasco, 2011*; *Roelfsema et al., 1998*; *Anton-Erxleben and Carrasco, 2013*).

Neuronal responses modulate as attention shifts among stimuli at different receptive-field locations (*Moran and Desimone, 1985*; *Treue and Maunsell, 1996*; *Reynolds et al., 1999*; *Martínez-Trujillo and Treue, 2002*; *Ghose and Maunsell, 2008*; *Lee and Maunsell, 2010*; *Ni et al., 2012*; *Recanzone and Wurtz, 2000*; *Luck et al., 1997*; *Motter, 1993*; *Chelazzi et al., 1998*; *Zénon and Krauzlis, 2012*). These response modulations can be complicated. For example, depending on the stimulus configuration, attending to a non-preferred stimulus can either increase or suppress activity (e.g. *Treue and Maunsell, 1996*). Normalization models of attention provide a succinct framework in which these complex response modulations can be understood (*Reynolds et al., 1999*; *Ghose, 2009*; *Reynolds and Heeger, 2009*; *Lee, 2009*; *Boynton, 2009*). However, only a few studies have directly tested these models against the responses of individual neurons to various stimulus configurations (*Ni et al., 2012*; *Lee, 2009*; *Sanayei et al., 2015*; *Xiao et al., 2014*).

Normalization models of attention assume that attention acts on stimulus-induced excitation and suppression to modulate neuronal responses. Importantly, both excitation and suppression vary

**eLife digest** At any moment, our brain receives an enormous amount of information from our senses. However, we are not aware of all of this information; only the information we decide to focus on is perceived in detail. This ability to focus our attention is important for survival.

The neurons involved in vision respond best to information that comes from a small 'window' in what is being seen. When something appears in this window (known as the neuron's receptive field), the activity of the neuron either increases or decreases. How does focusing attention on an object change the neuron's response? Verhoef and Maunsell investigated this question by recording electrical activity in an area of the brain called V4 in monkeys as they focused their attention on objects in different locations of the neuron's receptive field.

The recordings show that a single rule determines when attention influences a neuron's activity. If an object inside the neuron's receptive field decreases the activity of the neuron, then attention can change that neuron's activity. Attention then changes the activity of the neuron by either removing or further boosting the influence of these objects.

Verhoef and Maunsell then developed a mathematical model based on these results, and found that the model could explain why the activity of a neuron changes when attention is focused on objects at different locations in its receptive field. The next step is to understand exactly how the brain works to either remove or boost the influence of an object that causes a neuron's activity to decrease.

spatially within the receptive field: excitation is largely restricted to the classical receptive field (cRF), while suppression extends far beyond into the surround (*Cavanaugh et al., 2002a*, *2002b*; *Sceniak et al., 1999*; *Desimone and Schein, 1987*; *Carandini et al., 1997*). Crucially, how attention interacts with the receptive field structure of neurons remains unclear. For example, the way that attention acts on neuronal responses when shifted among stimuli inside the cRF versus when shifted to stimuli inside the surround has not been compared directly (*Motter, 1993*; *Sanayei et al., 2015*; *Sundberg et al., 2009*). Differences may occur because feedforward-, feedback- and intracortical circuitries are thought to contribute differentially to the suppressive and excitatory inputs associated with stimuli in either the cRF or the surround (*Angelucci et al., 2014*), and because the cRF and the surround presumably serve different functional roles (*Angelucci et al., 2014*; *Schwartz and Simoncelli, 2001*; *Vinje and Gallant, 2000*). More generally, it remains unknown if and how attention operates on the spatially-varying excitation and suppression of a neuron's receptive field. This is a pivotal open question because, as we will show below, the interaction between attention and the receptive field structure determines which neurons are most affected by attention and consequently are most likely to influence attentional behavior.

We measured how attention affects neuronal responses to various stimulus configurations both inside and outside the cRF of V4 neurons, and fitted normalization models to the responses of individual neurons. We find that the principles that drive attention modulation are remarkably similar within the classical receptive field and the surround. We show that stimuli induce excitation and suppression that varies spatially, and that attention interacts with this spatially-varying excitation and suppression. This interaction explained the large differences in attention modulations across neurons, and a non-additive relationship among stimulus selectivity, stimulus-induced suppression and attention modulation. A spatially-tuned normalization model, wherein attention multiplies both the excitatory and spatially-varying normalization term, precisely accounted for all neuronal responses to either single or multiple stimuli, either attended or unattended, presented inside either the cRF or the surround. The model relates stimulus selectivity, stimulus-induced suppression and attention-related modulation to each other, and unifies spatial summation and attention-related modulation across different regions of the receptive field.

# Results

## Task and behavioral performance

We trained two rhesus monkeys to perform a visual-detection task in which spatial attention was controlled and measured. In each trial, a sequence of stimuli was presented at four locations equidistant from the fixation point (*Figure 1A*). Stimuli were full-contrast static Gabor stimuli with one of two orthogonal orientations. The monkey's task was to detect a faint white spot (target; *Figure 1A* right) that appeared in the center of one Gabor during a randomly selected stimulus presentation. We manipulated attention in blocks of trials by cueing the monkey at the start of each block as to which stimulus location was most likely to contain the target (Materials and methods). In 91% of trials the target was presented at the cued location (valid cue; position of the black circle *Figure 1A*). On the remaining 9% of trials the target appeared with equal probability at one of the three uncued locations: either next to the cued location (invalid near; position of the yellow circle *Figure 1A*), or at one of two locations contralateral to the cued location (invalid far; position of the blue circles *Figure 1A*).

The attention cue considerably affected behavioral performance in the task: targets were much more likely detected at a cued location than at an uncued location, even when the uncued location was adjacent to the cued location (*Figure 1B,C*; valid vs. invalid near: monkey M1: $p=8 \times 10^{-27}$, M2: $p=1 \times 10^{-26}$; valid vs. invalid far: M1: $p=1 \times 10^{-36}$, M2: $p=2 \times 10^{-58}$; paired t-test on the average proportion correct across sessions; M1: N = 52; M2: N = 78). The improved performance indicates that the monkeys preferentially attended to the cued stimulus location, which allowed us to compare neuronal responses among conditions in which attention was directed to different stimulus locations within neurons' cRF or surround.

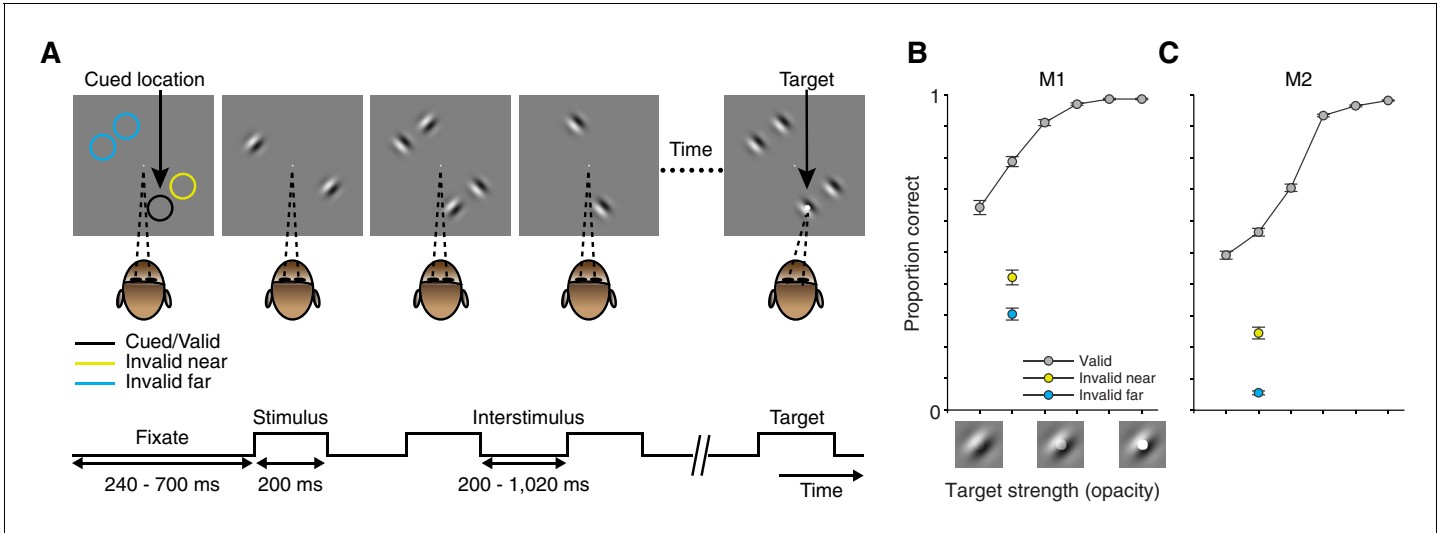

**Figure 1.** Task and performance. (**A**) Every trial consisted of a sequence of stimulus presentations. On each stimulus presentation (200 ms duration; 200–1020 inter-stimulus interval), Gabor stimuli of two orthogonal orientations could be presented at four possible stimulus locations. The monkey was rewarded for detecting a faint white spot (target) in the center of one Gabor during one stimulus presentation. For 91% of trials the target was presented at the cued location (location of the black circle; valid trials). On the remaining 9% of trials the target was presented at one of three uncued locations: adjacent to the cued location (location of the yellow circle; invalid near), or at one of two locations on the opposite side of the fixation point (location of the blue circles; invalid far). Colored circles in (**A**) are shown for illustrative purposes, never presented during the task. (**B**) Average performance across recording sessions for monkey M1. Proportion correct (± SEM based on N = 52 sessions; proportion correct at equal target strength: Valid: 0.79; Invalid near: 0.42; Invalid far: 0.30) as a function of target strength for trials in which the target occurred at the cued (gray: valid) or uncued (yellow: invalid near; blue: invalid far) location. Target strength is defined as the opacity of the target. The pictograms below the target-strength axis illustrate the nature of the target-strength manipulation but do not represent actual target-strength values used during the recordings. (**C**) Average performance across recording sessions for monkey M2 (N = 78 sessions; proportion correct at equal target strength: Valid: 0.56; Invalid near: 0.24; Invalid far: 0.05).

## Experimental conditions and example neurons

We examined the principles by which attention affects neuronal responses to stimuli inside the classical receptive field (cRF) or within the surround (sRF). Using chronically implanted microelectrode arrays, we recorded from 728 neurons in visual area V4 in the left hemisphere of two monkeys (monkey M1: 264; M2: 464) while they performed the visual-detection task in which spatial attention was controlled. All results presented here are based on the activity of these 728 single neurons, but all findings were confirmed in the responses of 12,067 multi-unit clusters (M1: 4709; M2: 7358). During each session we simultaneously measured the activity of multiple neurons, and optimized the orientation and position of stimuli for a randomly selected unit. The neurons' receptive field centers were located in the lower right visual field (black dots in *Figure 2A* for an example session).

In different blocks of trials, we measured neuronal responses to stimuli presented at three different receptive-field locations (stimulus locations 1, 2, 3 in *Figure 2A*). Within a block of trials, only two of these stimulus locations were used: e.g. location 1+2 or 1+3 in *Figure 2A* (Materials and methods). During each stimulus presentation within a trial, we presented one, two, or no stimuli at the two stimulus locations near the receptive field (*Figure 2B*).

Depending on the location of each neuron's receptive field, stimuli fell either inside the cRF or within the surround. We distinguished between two receptive-field configurations: one in which the two stimulus locations both lay inside the neuron's cRF (cRF-cRF, *Figure 2C*), and another in which one stimulus location was positioned inside the neuron's cRF while the other stimulus location was positioned inside its surround (sRF-cRF, *Figure 2D*). Because we tested the responses to stimuli shown at two locations pairings (e.g. locations 1+2 in *Figure 2C* vs. 1+3 in *Figure 2D*), 309 neurons were tested in both a cRF-cRF and an sRF-cRF configuration (M1: 97; M2: 212). We classified locations as belonging to the cRF or sRF using stimulus presentations that included only one Gabor (*Figure 2B*; Materials and methods). Locations where either stimulus orientation generated a response were considered to lie within the cRF. Those where neither stimulus orientation generated a response were considered to lie within the surround (*Figure 2—figure supplement 1*).

In different blocks of trials, the monkeys directed their attention toward all possible stimulus locations, one attended location per block of trials, each time ignoring the other stimulus locations. Attention was directed toward stimulus locations near the neurons' receptive fields (e.g. locations 1, 2, or 3 in *Figure 2A*), or toward stimulus locations away from the receptive fields ('Away' in *Figure 2C,D*; *attend away*), i.e. to stimulus locations on the opposite side of the fixation point from the neuron's receptive field.

We quantified the stimulus selectivity of the neurons separately for each stimulus configuration. For each of four Gabor pairs (*Figure 2B*) at each pair of stimulus locations (i.e. location pairings 1+2 or 1+3, *Figure 2A*), we used a selectivity index ('Selectivity', *Figure 2E*): $(P−N)/(P+N)$, that ranges from zero (unselective) to one (completely selective). Here $P$ is the response to the component Gabor of a Gabor pair that generated the stronger average response when presented alone (preferred), and $N$ is the response to other component Gabor that generated the weaker average response when presented alone (non-preferred). Note that the preferred and non-preferred Gabor within a pair were presented at two different receptive-field locations, and could have the same or a different orientation (*Figure 2B*). Thus stimulus selectivity between members of a Gabor pair could arise from a neuron's orientation selectivity and from its preference for spatial locations. In subsequent analyses, we will show that the relationship between attention modulation and stimulus selectivity does not depend on whether the stimulus feature is space or orientation. What is critical for attention modulation is a differential response to the component stimuli of a compound stimulus.

For both the cRF-cRF and sRF-cRF condition, we measured each neuron's stimulus-induced suppression for each Gabor pair at each pair of stimulus locations using a stimulus-induced suppression index: $(P−PN)/(P+PN)$ (middle right pictogram *Figure 2E*). $PN$ is the response to the Gabor pair (P and N defined as before). This index is negative when the neuronal response increases when a non-preferred stimulus is added to the preferred stimulus, and positive when the neuronal response is suppressed by the addition of a non-preferred stimulus to the preferred stimulus. By definition, neurons do not respond to a stimulus when it appears alone inside the surround, so the surround stimulus is invariably assigned as non-preferred (N).

For both the selectivity index and the stimulus-induced suppression index, the responses to the preferred (P), non-preferred (N), and their combined presentation (PN) were measured in the same

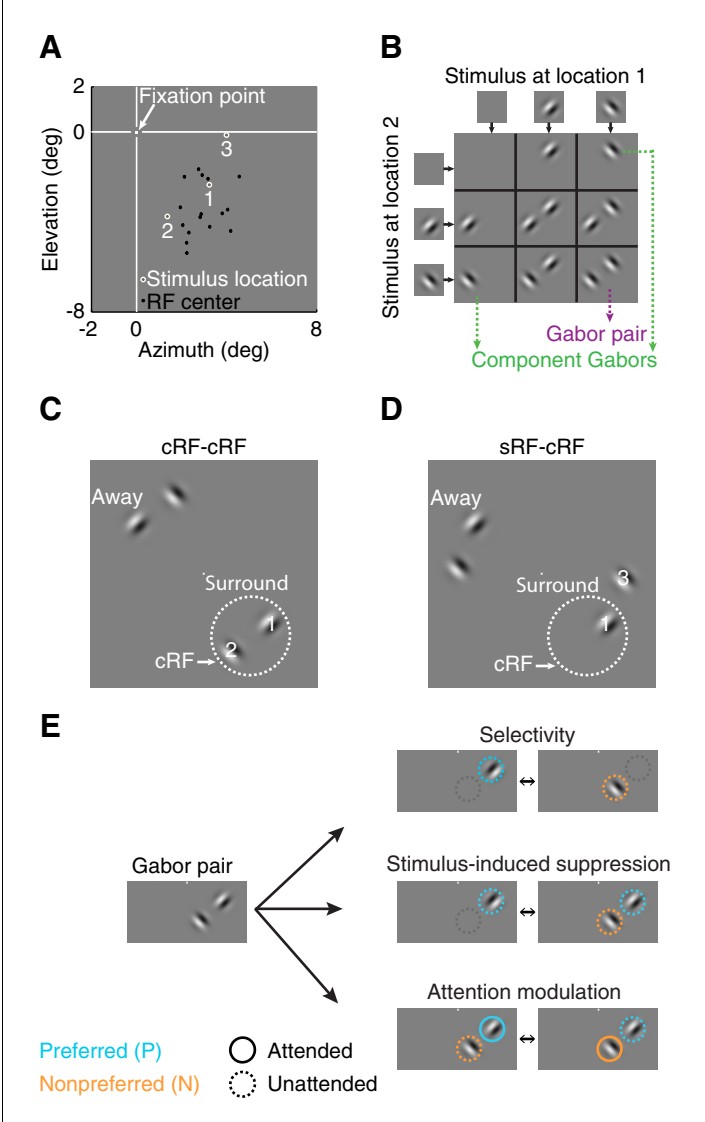

**Figure 2.** Stimulus conditions. (**A**) Neurons' receptive field centers were located in the lower right visual field: black dots indicate receptive-field centers of 16 simultaneously recorded neurons from one recording session. White circles (1,2,3) indicate the three stimulus locations near the neurons' receptive field for this example session. Within a block of trials, only two stimulus locations were used: locations 1+2 or 1+3. (**B**) Nine possible stimulus combinations resulting from two stimulus locations and two orthogonal orientations. (**C**) Two receptive-field configurations: cRF-cRF stimulus configuration with two stimuli inside the neuron's classical receptive field. White dotted circle illustrates the cRF. (**D**) sRF-cRF stimulus configuration with one stimulus inside a neuron's cRF and an adjacent stimulus in its surround. Each stimulus location near the neurons' receptive fields (stimulus location 1,2,3 in 2A) had a corresponding stimulus location on the opposite side of the fixation point (stimuli near *Away* in **C**, **D**; see also *Figure 2—figure supplement 1*). (**E**) Pictograms illustrate for one Gabor pair the stimulus configurations used to calculate all indices. Cyan circles indicate the preferred Gabor (P), orange circles the non-preferred Gabor (N). Solid circles represent task conditions wherein attention was directed toward a stimulus location near the neurons' receptive field ($P^{Att}N$, $PN^{Att}$). Dashed circles indicate that the stimulus was unattended and attention was directed toward another location.

The following figure supplement is available for figure 2:

**Figure supplement 1.** Average PSTH for individual Gabor stimuli presented inside the classical receptive field (cRF) and within the surround.

attention state: when attention was directed away from the neuron's receptive field (*attend away*). These responses are shown in the bar-plot insets in *Figure 3A–D*.

*Figure 3A–D* shows examples of attention-related response modulations of four different neurons to one selected Gabor pair: two neurons in the cRF-cRF configuration (**A, B**) and two neurons in the sRF-cRF configuration (**C, D**). The neuron in *Figure 3A* responded selectively to the two component Gabors of the Gabor pair shown inside the neuron's cRF (inset: P vs. N; selectivity index=0.44). Its response to the preferred Gabor was suppressed when the non-preferred Gabor was added to it (inset: P vs. PN; suppression index=0.22). The position of attention profoundly affected this neuron's responses: Compared to when attention was directed away from the neuron's receptive field (dashed green line; PN; *attend away*), attention to the preferred Gabor increased this neuron's response (cyan line; $P^{Att}N$), whereas attention to the non-preferred Gabor suppressed its response (orange line; $PN^{Att}$).

Attention-related modulation was quantified using an attention-modulation index: $(P^{Att}N - PN^{Att})$ / $(P^{Att}N + PN^{Att})$ (lower pictogram *Figure 2E*), which is positive when the neuronal response increases when attention is directed toward the preferred Gabor, compared to when attention is directed toward the non-preferred Gabor. The attention-modulation index for example 1 was 0.48.

In contrast to example 1, the response of the neuron in *Figure 3B* was poorly selective to the component Gabors of the Gabor pair (P vs. N; selectivity index=0.066), showed little suppression

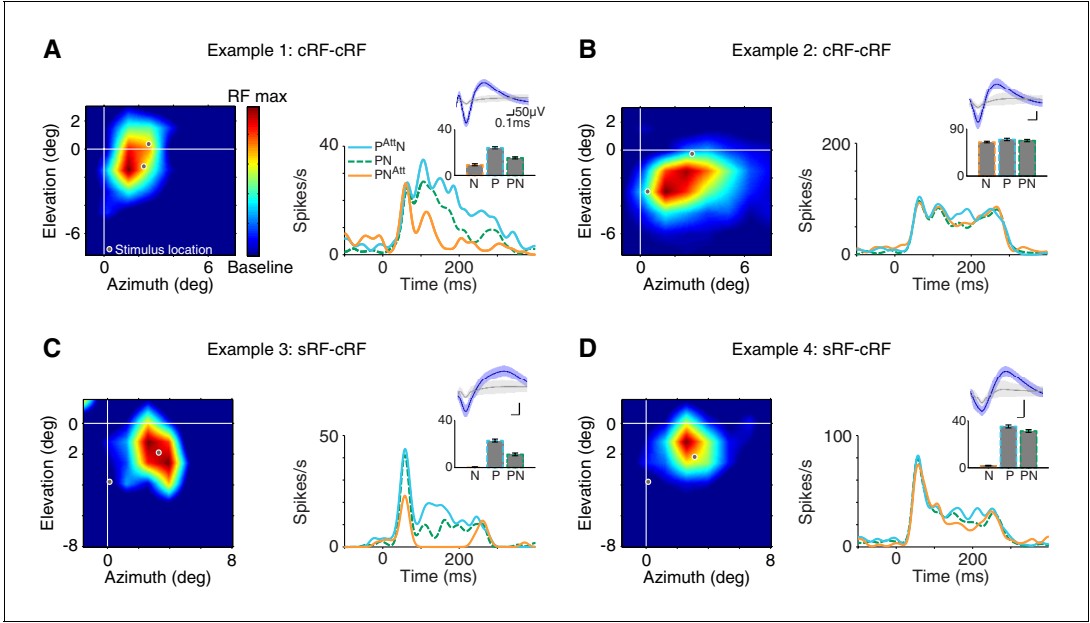

**Figure 3.** Example attention modulations. Responses of four different neurons to a selected Gabor pair are shown (measured in different sessions). (**A**) Example 1: cRF-cRF configuration. Left panel shows this neuron's receptive-field map with the two stimulus locations at which the Gabors were presented overlaid (white-gray dots). Right panel PSTHs show the neuronal responses to the Gabor pair when attention was directed toward the preferred Gabor (cyan line; $P^{Att}N$), the non-preferred Gabor (orange line; $PN^{Att}$), or a stimulus on the opposite side of the fixation point (green dashed line; PN; *attend away*). Bar-plot inset shows the responses of this neuron to a Gabor pair (PN) and its component Gabors (P, N), all measured in the *attend away* condition. This neuron's response was selective to the component Gabors of the Gabor pair (P vs. N), suppressed by the addition of a non-preferred Gabor to a preferred Gabor (P vs. PN), and strongly modulated when attention was shifted between the two component Gabors of the Gabor pair ($P^{Att}N$ vs. $PN^{Att}$). (**B**) Example 2: another neuron in the cRF-cRF configuration. This neuron showed weak selectivity, hardly any suppression, and little attention modulation. (**C**) Example 3: sRF-cRF configuration with one Gabor inside the neuron's cRF, and one Gabor inside its surround. By definition, the cRF Gabor is preferred (P) and the silent surround Gabor is non-preferred (N). The neuron responded highly selectively to the cRF and the surround Gabor when presented alone (P vs. N), showed surround suppression (P vs. PN), and was modulated by attention ($P^{Att}N$ vs. $PN^{Att}$). (**D**) Example 4: another neuron in the sRF-cRF configuration. This neuron was highly selective to the component Gabors of the Gabor pair, but only weakly suppressed by the surround Gabor, and showed little attention modulation. The insets show the average waveforms of the recorded neurons (blue) plus that of the multi-unit activity measured at the same electrode (grey). Shading around the mean represents ± 2 median absolute deviation (MAD). Scale bars indicate 50 µV and 0.1 ms. The receptive-field maps were normalized to the maximum response for each neuron during receptive-field mapping (RF max), dark blue shows the baseline response. Error bars represent ± SEM.

when a non-preferred Gabor was placed alongside a preferred Gabor (P vs. PN; suppression index=0.04), and was only weakly modulated when attention shifted between the preferred and the non-preferred Gabor within the cRF (cyan vs. orange line; attention-modulation index=0.04).

*Figure 3C* shows the responses of a neuron to a Gabor pair in another stimulus configuration, in which one Gabor was placed inside the neuron's cRF and another Gabor inside its surround (sRF-cRF). As expected, the neuron responded much more to the cRF Gabor than to the surround Gabor (P vs. N; selectivity=0.963). When the surround Gabor was placed alongside the cRF Gabor, the neuron's response was greatly reduced, the hallmark of surround suppression (P vs. PN; suppression index=0.336). The neuron showed strong attention-related modulation: Compared to when attention was removed from both the cRF and the surround Gabor (dashed green line; *attend away*), attention to the cRF Gabor increased this neuron's response (cyan line), while attention to the surround Gabor sharply decreased its response (orange line; attention-modulation index=0.58).

The response of the fourth example neuron in *Figure 3D* was highly selective to the component Gabors of the Gabor pair (P vs. N), only slightly suppressed by the surround Gabor (P vs. PN), and its firing rate was barely modulated by attention (cyan vs. orange line; selectivity index=0.9; suppression index=0.05; attention-modulation index=0.08).

These examples illustrate the diverse stimulus selectivities, stimulus interactions (i.e. stimulus-induced suppression) and attention-related modulations in the neuronal responses in visual cortex. Next, we asked how variability in stimulus selectivity and stimulus-induced suppression relates to variability in attention modulation within the cRF and the surround across the sample of recorded neurons.

## Relationship among selectivity, stimulus-induced suppression and attention modulation

We first examined the relationship between selectivity and attention modulation. Shifting attention between two Gabors inside the cRF was associated with larger response changes for neurons with more selective responses to the component Gabors of the Gabor pair (*Figure 4A*; cRF-cRF configuration; p=4 $\times$ $10^{-109}$ for a non-zero slope; linear regression) (*Reynolds et al., 1999*). Attention-related modulation was also stronger for neurons that responded more selectively to the cRF and surround stimulus (*Figure 4B*; sRF-cRF configuration; p=3 $\times$ $10^{-76}$ for a non-zero slope; linear regression). Low selectivity can occur in the sRF-cRF configuration when the cRF stimulus produces little response because it has a non-preferred orientation or is positioned at a weakly responsive cRF location. Comparing *Figure 4A and B* shows that attention-related modulation increases more with selectivity in the cRF-cRF than in the sRF-cRF configuration (p=5 $\times$ $10^{-4}$ for different slopes in each receptive-field configuration; general linear model).

We next examined stimulus-induced suppression. V4 neuronal responses on average decrease when a non-preferred stimulus is added to a preferred stimulus inside their cRF (*Figure 4C*; average suppression index = 0.08, p=4 $\times$ $10^{-104}$; t-test for a difference from zero) (*Reynolds et al., 1999*). Similarly, stimulating the surround decreases the average responses of V4 neurons (*Figure 4D*; average suppression index = 0.04, p=2 $\times$ $10^{-28}$; t-test) (*Schein and Desimone, 1990*). However, stimuli inside the surround suppressed the neuronal responses less than stimuli inside the cRF: the average suppression index for the surround condition (sRF-cRF) was significantly smaller than the average suppression index for the cRF condition (M1: p=9 $\times$ $10^{-6}$; M2: p=4 $\times$ $10^{-15}$; t-test; see below and Figure 7 for further discussion). The black bars in *Figure 4C and D* represent neurons that were significantly (p<0.01) suppressed by the non-preferred (surround) stimulus. See *Figure 4—figure supplement 1* for some example neurons with significant surround suppression (see also *Figure 3C*). Surround suppression was also weaker than cRF suppression when comparing only suppression indices that differed significantly from zero (p<0.001).

Extending previous findings in area MT (*Ni et al., 2012*; *Lee, 2009*), we find that V4 neurons with stronger stimulus-induced suppression by cRF stimuli also showed stronger attention modulation (*Figure 4E*; p=1 $\times$ $10^{-31}$ for a non-zero slope; linear regression). Furthermore, and consistent with a previous study (*Sundberg et al., 2009*), attention modulation was also stronger for neurons whose responses were more suppressed by a surround stimulus (*Figure 4F*; p=5 $\times$ $10^{-4}$ for a non-zero slope; linear regression). However, comparing *Figure 4E and F* shows that attention-related modulation increases more with stimulus-induced suppression in the sRF-cRF than in the cRF-cRF configuration (p=0.005 for different slopes in each receptive-field configuration; general linear model).

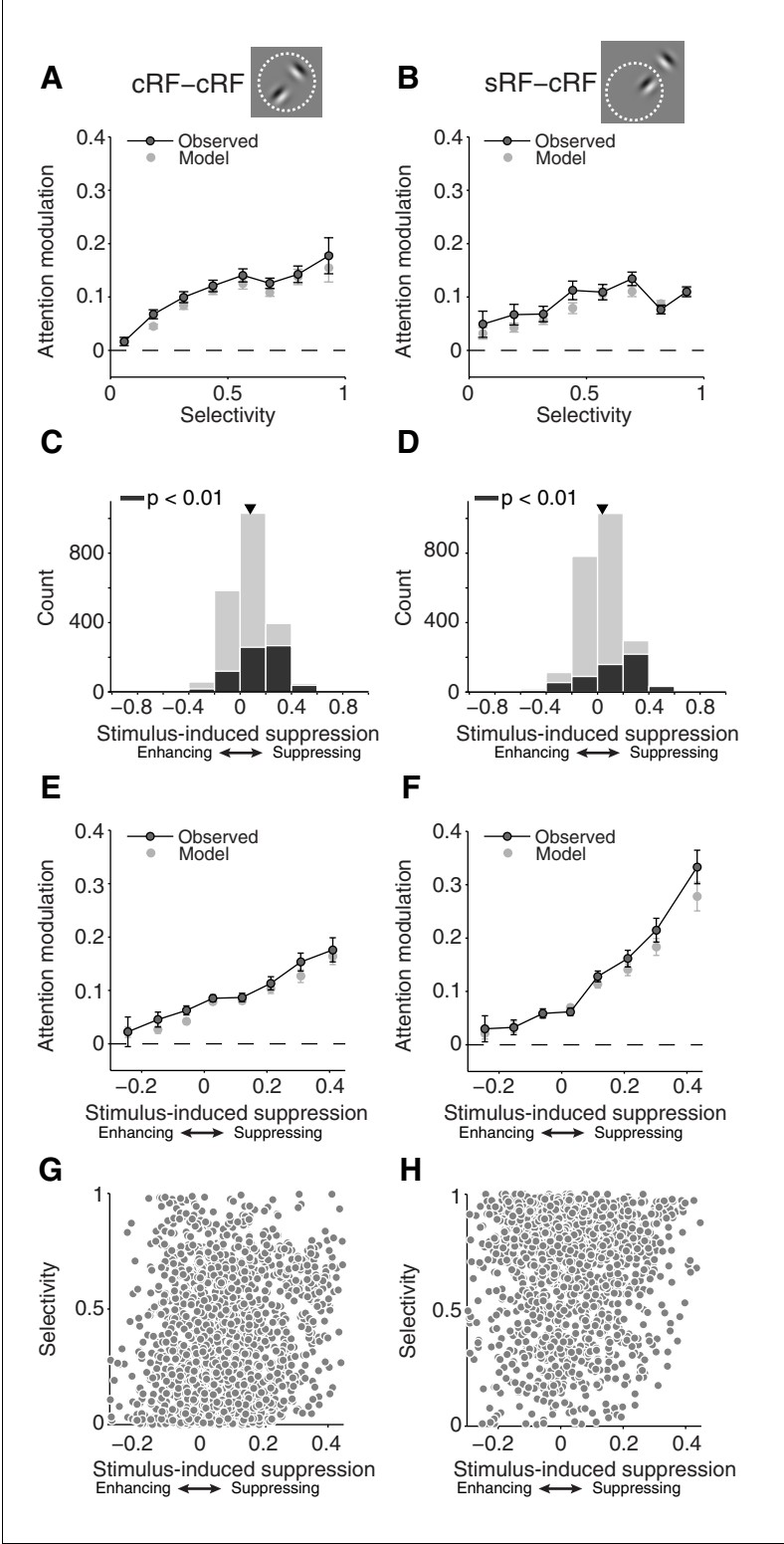

**Figure 4.** First-order analyses suggest that attention modulation follows different principles for stimuli inside the cRF and the surround. (**A, B**) Average attention modulation as a function of the stimulus selectivity in the cRF-cRF and sRF-cRF configuration respectively. Low selectivity occurs in the sRF-cRF configuration when neurons respond weakly to the cRF stimulus, e.g. because of a non-preferred orientation or a weakly responsive cRF location, and have a baseline response to the surround stimulus. (**C, D**) Histogram of all stimulus-induced suppression indices measured in the cRF-cRF and sRF-cRF configuration respectively. The suppression index is negative when neurons

*Figure 4 continued on next page*

*Figure 4 continued*

increase their response when a non-preferred stimulus is added to the preferred stimulus (*enhancing*), and positive when neurons decrease their response when a non-preferred stimulus is added to the preferred stimulus (*suppressing*). Black bars indicate indices associated with Gabor pairs for which the suppression index differed significantly from zero (p<0.01; permutation t-test; see also *Figure 4—figure supplement 1*). Triangle points to the mean suppression index. (E, F) Average attention modulation as a function of stimulus-induced suppression in the cRF-cRF and sRF-cRF configuration respectively. Error bars represent ± SEM. (G, H) Stimulus-induced suppression versus stimulus selectivity for all Gabor pairs in the cRF-cRF (N = 1769) and sRF-cRF (N = 1768) configuration respectively.
The following figure supplement is available for figure 4:

**Figure supplement 1.** Example neurons with strong surround suppression.

A previous study in V4 examining the relationship among stimulus selectivity, sensory interaction (akin to stimulus-induced suppression) and attention modulation, found a strong correlation between stimulus selectivity and sensory interaction (*Reynolds et al., 1999*). In the present study, however, stimulus selectivity and stimulus-induced suppression were *not* significantly correlated with each other across neurons (*Figure 4G,H* Pearson correlation = 0.02, p=0.32; see Discussion for further comments on the difference between studies). This finding shows that the correlations between stimulus selectivity and attention modulation, and that between stimulus-induced suppression and attention modulation, are not explained by an underlying correlation between selectivity and suppression. Furthermore, and in contrast to previous studies, the lack of a correlation between both indices allowed us to examine the separate contributions of selectivity and suppression to the magnitude of attention modulation.

The above-mentioned different relationships between stimulus selectivity, stimulus-induced suppression and attention-related modulation in the cRF-cRF and the sRF-cRF configuration, suggest that the rules that govern attention modulation differ within the cRF and the surround. Next we falsify this suggestion and show how these different relationships in the two receptive-field configurations can be explained by a common rule.

## Stimulus selectivity and stimulus-induced suppression interact in determining attention modulation and do so similarly inside the cRF and the surround

We used multiple linear regression to examine if attention-related modulation depends on the joint magnitude of stimulus selectivity and stimulus-induced suppression. For both receptive-field configurations, the regression model included a main effect of selectivity and a main effect of stimulus-induced suppression. Importantly, in each RF configuration the regression model also included an interactive product term, which measured the dependency of attention-related modulation on both selectivity and stimulus-induced suppression, i.e. this term measures whether the relationship among selectivity, suppression and attention modulation is non-additive (see Materials and methods for further information).

*Figure 5* shows how attention modulation varies with selectivity and stimulus-induced suppression (**5A;** cRF-cRF, **5B;** sRF-cRF). For both configurations, the relationship is non-additive. Specifically, *Figure 5A and B* show that when stimulus-induced suppression is low, attention modulation will be weak, even when attention is shifted between a strong and a weak stimulus (upper left corner in *Figure 5A,B*). That is, the plots show that the effect of selectivity near zero stimulus-induced suppression is weak, although significant (main effect of selectivity at zero stimulus-induced suppression: cRF-cRF: p=$2 \times 10^{-64}$; sRF-cRF: p=$2 \times 10^{-60}$; M1: p=$7 \times 10^{-136}$ across RF configurations; M2: p=$5 \times 10^{-30}$ across RF configurations).

Conversely, when selectivity is low, attention modulation will also be weak, even when attention is shifted between stimuli that strongly suppress each other's response (bottom right corner in *Figure 5A,B*; effect of stimulus-induced suppression at zero selectivity: cRF-cRF: p=0.5; sRF-cRF: p=0.6; M1: p=0.4 across RF configurations; M2: p=0.3 across RF configurations).

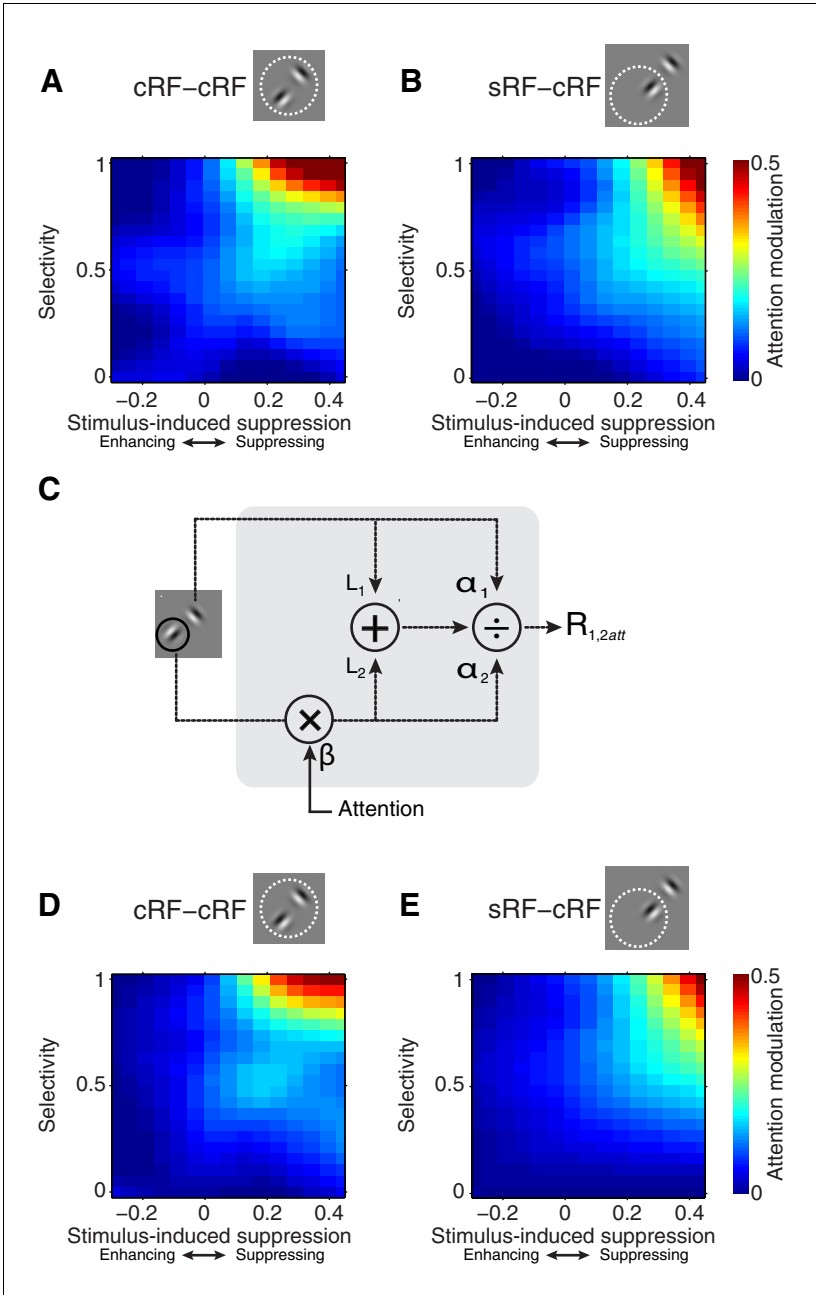

**Figure 5.** Selectivity and stimulus-induced suppression interact to control attention modulation. (**A**, **B**) Average attention modulation as a function of stimulus-induced suppression (x-axis) and stimulus selectivity (y-axis) in the cRF-cRF and sRF-cRF configuration respectively. The magnitude of attention modulation is indicated by color (red = strong, blue = weak). Note that, although the data covered most of this space (see *Figure 4G,H*), few regions, e.g. the lower right corner in (**B**), were not well sampled. (**C**) Model schematic. Every stimulus contributes an excitatory drive ($L_1$ and $L_2$) to the neuron's response ($R_{1,2att}$) to a Gabor pair. Each stimulated receptive-field location, either inside the cRF or inside the surround, contributes divisive suppression ($\alpha_1$ and $\alpha_2$) to the neuron's response. The divisive suppression is fixed for each receptive-field location, independent of the stimulus presented at that location. A small amount of baseline suppression is further added ($\sigma$ parameter; not shown). Directing attention toward a stimulus location has a multiplicative effect ($\beta$) on the parameters ($L_2$ and $\alpha_2$) corresponding to the attended receptive-field location (location 2 in the schematic). (**D**, **E**) Average model-predicted attention modulation as a function of the observed stimulus-induced suppression (x-axis) and the observed stimulus selectivity (y-axis) in the cRF-cRF and sRF-cRF configuration respectively (See also *Figure 5—figure supplement 1*). Same conventions as in (**A**, **B**).

*Figure 5 continued on next page*

*Figure 5 continued*

The following figure supplement is available for figure 5:

**Figure supplement 1.** Example single-neuron responses and their corresponding model fits.

Strong attention-related modulation occurs only when selectivity and suppression are both large, and this was true for both RF configurations (upper right corner in *Figure 5A,B*; interaction between stimulus-induced suppression and selectivity: cRF-cRF: $p=2 \times 10^{-9}$; sRF-cRF: $p=3 \times 10^{-6}$; M1: $p=1 \times 10^{-4}$ across RF configurations; M2: $p=3 \times 10^{-20}$ across RF configurations).

The interaction between selectivity and stimulus-induced suppression did not differ significantly between the cRF-cRF and sRF-cRF configuration (M1: $p=0.6$; M2: $p=0.85$; 3-way interaction), nor did any other interaction with RF configuration. Because a non-significant effect does not indicate the absence of an effect, we performed a Bayesian regression analysis (Materials and methods). This analysis showed that the observed data are 347 times more likely to agree with a regression model that does not distinguish between the cRF-cRF and sRF-cRF configurations than with a model that does include RF-configuration as a predictor. Thus attention modulation is driven by similar mechanisms within the cRF and the surround.

## Attention modulation depends on a general definition of stimulus selectivity

To examine response modulations associated with shifting attention between two receptive-field stimuli, previous studies used two different stimuli (e.g. stimuli of different orientations, colors, directions), each presented at a different but approximately equally-responsive cRF position (*Moran and Desimone, 1985*; *Reynolds et al., 1999*; *Ghose and Maunsell, 2008*; *Lee and Maunsell, 2010*; *Ni et al., 2012*). Similar to these previous studies, we reproduced the above findings using the data from conditions with low spatial selectivity. When attention shifted between stimuli at two approximately-equally responsive cRF positions (less than 2 spike/s response difference when each of two cRF positions is stimulated with an identical single stimulus), similar effects were observed (main effect of orientation selectivity: $p<0.001$; main effect suppression: $p=0.7$; interaction between feature selectivity and suppression: $p=0.02$; multiple linear regression).

Next, we examined whether the converse situation, i.e. same stimuli at unequally-responsive cRF positions, would produce attention modulations comparable to those described earlier. We found similar attention-related modulations using Gabor pairs consisting of identical Gabor stimuli presented at various cRF positions. Note that in this situation selectivity to the component Gabors of a Gabor pair originates solely from a neuron's spatial preferences (receptive field), because the Gabors are identical. All of the above findings were replicated using only the data obtained with Gabor pairs consisting of identical Gabors (main effect spatial selectivity: $p=4 \times 10^{-22}$; main effect suppression: $p=0.29$; interaction between spatial selectivity and suppression: $p=6 \times 10^{-6}$; multiple linear regression). This indicates that attention-related modulation depends on a differential response to the component stimuli of a compound stimulus, regardless of the origin of the response difference (feature or spatial). Accordingly, at their most abstract level, models of neuronal attention modulation only need to account for the responses arising from different component stimuli, whether they arise from preferences for specific stimulus features, preferences for certain parts of the receptive field, or both.

## A spatially-tuned normalization model captures attention modulation inside the cRF and in the surround

Our findings reveal a striking uniformity in the rules that govern attention modulation inside the cRF and within the surround: the interaction between stimulus selectivity and stimulus-induced suppression strongly influences how much attention modulates neuronal responses. Hence, any model of neuronal attention modulation needs to embody this relationship. We found that a spatially-tuned normalization model can readily capture this interaction (Materials and methods).

We used a spatially-tuned normalization model, described as follows (*Figure 5C*):

$$R_{1,2} = \frac{L_1 + L_2}{\alpha_1 + \alpha_2 + \sigma} \qquad (1)$$

where $R_{1,2}$ is the neuronal response to a Gabor pair consisting of Gabors 1 and 2. $L_1$ and $L_2$ are the excitatory drives associated with each component Gabor. The $\alpha_1$ and $\alpha_2$ parameters control the suppressive drive of each stimulated cRF or surround location. In this model, $\alpha_1$ and $\alpha_2$ are each associated with one receptive-field location, and do not vary with the orientation of the stimuli shown at those locations. Because the suppression, or normalization, is free to vary across receptive-field locations, the normalization is spatially tuned. In fitting the data, we fixed $\alpha_1$ at one to constrain the model. The $\alpha$ parameter adds baseline suppression. Directing attention toward the first ($R_{1att,2}$; *Equation (2)*) or second ($R_{1,2att}$; *Equation (3)*) receptive-field location has a multiplicative effect on the parameters corresponding to the attended receptive-field location. This is described by the $\beta$ parameter in *Equations (2) and (3)*:

$$R_{1att,2} = \frac{\beta L_1 + L_2}{\beta + \alpha_2 + \sigma} \qquad (2)$$

$$R_{1,2att} = \frac{L_1 + \beta L_2}{1 + \beta \alpha_2 + \sigma} \qquad (3)$$

The model was fit to each neuron's responses in all stimulus conditions: including conditions with one stimulus or two stimuli near the receptive field, and conditions with attention directed toward stimulus locations near the receptive field or away from it (Materials and methods).

The spatially-tuned normalization model provided an accurate account of the neuronal data, giving a median two-fold cross-validated explained variance of 87% across neurons (M1: 86%; M2: 88%). For the 309 neurons (M1: 97; M2: 212) that were tested in both a cRF-cRF and an sRF-cRF configuration the responses were equally well explained (M1: 86%; M2: 87%).

The model captures the way attention modulates neuronal responses to stimuli inside the cRF or the surround. *Figure 4A,B,E,F* (light grey points) show that the model precisely accounts for attention modulation across the full range of observed stimulus selectivity and stimulus-induced suppression values, within both the cRF-cRF and sRF-cRF configuration. *Figure 5D,E* shows the average model predictions based on the model fits from all neurons in the cRF-cRF (**D**) and sRF-cRF (**E**) configurations (see *Figure 5—figure supplement 1* for response fits of individual neurons). The model reproduces the way stimulus selectivity and stimulus-induced suppression interacted in both the cRF-cRF and the sRF-cRF configurations: predicting large attention modulation when both selectivity and suppression are strong, but little attention modulation when either selectivity or suppression is low. Thus this single model describes how attention modulates responses to stimuli inside the cRF or the surround.

The previous analyses were based on stimulus configurations with two stimuli inside the neurons' receptive field. Importantly, the model also accounts for the neuronal effects of attention to single stimuli at different receptive field locations. This is shown in *Figure 6*, which shows how the effect of attention varies when attention is directed to single stimuli at various distances from the receptive field center: for the observed (**A**) and the modeled (**B**) responses. Attention modulation to single stimuli is typically small compared to attention modulation with multiple stimuli. This is because in single stimulus conditions there are no suppressive influences from a flanking stimulus, and we have shown that these suppressive influences are necessary to induce strong attention modulation (*Figure 5*). The model accounts for these smaller attention modulations with single stimuli.

## Stimulus-induced suppression depends on distance from the receptive-field center

What determines the spatial tuning of suppression? For each neuron we sorted the value of the suppression-parameter $\alpha$, associated with each of the three measured receptive-field locations (*Figure 2A*), as a function of the distance of each receptive-field location from the neuron's receptive-field center (*Figure 7A*). Within neurons, locations closest to the receptive-field center induced on average greater suppression than locations furthest away from the receptive-field center (average $\alpha$-parameter values of closest vs. furthest location: p=$2 \times 10^{-23}$; M1: p=$7 \times 10^{-5}$; M2: p=0.005;

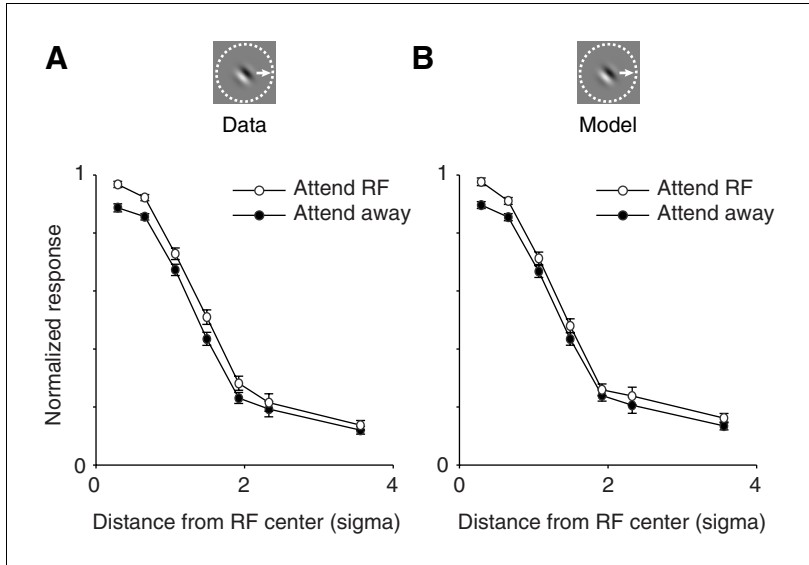

**Figure 6.** The spatially-tuned normalization model captures how attention modulates responses to single stimuli presented at various receptive field locations. (**A**) Observed responses. Average response as a function of the distance between the single stimulus and the receptive field center, when the stimulus is attended (white) or unattended (black). (**B**) Same as (**A**) but for the modeled responses. The responses of each neuron were normalized to the maximum response across conditions in which a single stimulus was presented inside the receptive field. The receptive field distance is given by the Mahalanobis distance from the Gaussian receptive-field center. The Mahalanobis distance is akin to the number of standard deviations ($\sigma$) away from the receptive-field center. Only neurons whose receptive fields were well fitted with a two-dimensional Gaussian profile (>80% explained variance; 306 neurons; M1: 95; M2: 211) were included. Error bars represent ± SEM.

paired permutation t-test). This is further illustrated in *Figure 7B,C* (gray), which shows the average normalized $\alpha$-parameter value as a function of the distance from the receptive-field center. Lower $\alpha$-values at greater distances are consistent with the observation that surround suppression is significantly weaker than cRF suppression (*Figure 4C,D*; see above).

In addition to suppression, *Figure 7B,C* also shows the strength of the excitatory drive (measured by the average *L*-parameter values) as a function of the distance from the receptive-field center (black). Comparing the curves for the excitatory (black) and the suppressive drive (gray) reveals a striking similarity between the receptive field structure of V4 neurons and that previously observed in primary visual cortex (V1) (*Cavanaugh et al., 2002b*; *Sceniak et al., 1999*; *DeAngelis et al., 1994*): both excitation and suppression are maximal near the receptive-field center, but excitation is more spatially concentrated, with suppression stretching over larger distances.

The spatial pattern of suppression suggests that suppression in the cRF and the surround are continuous extensions of one another. These findings indicate that attention operates uniformly on the spatially-continuous excitation and suppression of a neuron's receptive field.

## Spatial variability in excitation and suppression underlies differences in attention modulation across neurons

Is a variable top-down attention signal necessary? We fixed the attention parameter $\beta$ at a constant value for all neurons, i.e. as the mean $\beta$ value across neurons when estimated as a free parameter. This constrained model accounted almost as well for the data as the model in which $\beta$ was free to vary (median two-fold cross-validated percentage explained variance 86%, compared to 87% for the unconstrained model). This finding suggests that differences in attention modulation between neurons are only weakly related to differences in the top-down attention signal across neurons (see also [*Ni et al., 2012*]).

In contrast, spatial tuning is important. When we instead kept the $\alpha$ terms constant and allowed $\beta$ to vary across neurons, the model's performance decreased significantly (77%; p=0.004, Sequential

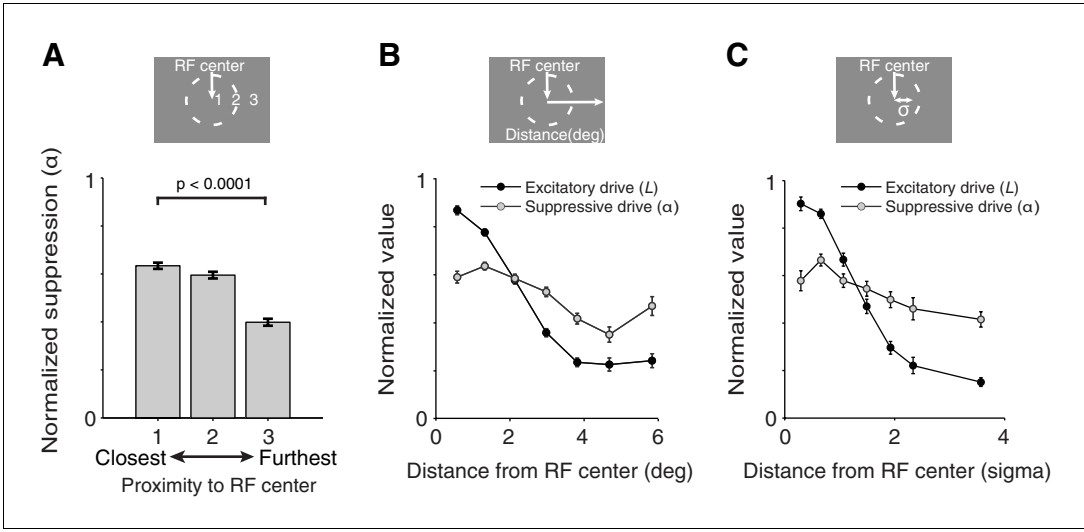

**Figure 7.** Spatially-tuned excitation and suppression decrease with distance from the receptive-field center, but at different rates. (A) Each recording session we measured neuronal responses to stimuli presented at three different receptive-field locations (*Figure 2A*). The responses of each neuron were fitted with the spatially-tuned normalization model. The value of the suppression parameters $\alpha$ associated with each of the three measured receptive-field locations were ranked according to the proximity of those receptive-field locations to the neuron's receptive-field center: 1 being closest, and 3 being furthest away from the receptive-field center. The suppression parameter values were then normalized by the maximum $\alpha$-value for each neuron. For each ranking number, the normalized suppression parameter values were subsequently averaged across neurons. Stimulus locations closest to the receptive-field center contributed more suppression to the neurons' response than those furthest away. (B) Average normalized suppressive drive ($\alpha$, gray) and excitatory drive ($L$, black) as a function of the distance (in visual degrees) of its corresponding receptive-field location from the receptive-field center. The value of the excitatory drive parameter $L$ for stimuli of different orientations were averaged per receptive-field location, and normalized by the maximum excitatory drive across the three measured receptive-field locations of a neuron. (C) Same as (B) but with an alternative distance measure, namely the Mahalanobis distance, which is akin to the number of standard deviations ($\sigma$) away from the receptive-field center. In (B) and (C), each excitatory-drive value $L$ (black) has a corresponding suppressive-drive value $\alpha$ (gray). Error bars represent ± SEM.

F-test), especially for neurons that were tested in both a cRF-cRF and an sRF-cRF configuration (73%; p=0.002).

Hence when a fixed stimulus is shown, differences in attention modulation across neurons appear to arise when a relatively uniform top-down attention signal interacts with the different amounts of excitation and suppression elicited by that stimulus in each neuron.

## Discussion

We measured the dependency of neuronal attention modulation on stimulus selectivity and stimulus-induced suppression throughout different receptive field regions, including the surround, of V4 neurons. We found that stimulus selectivity and stimulus-induced suppression strongly interact to determine the magnitude of attention modulation in neurons. This interaction determined attention modulation within both the classical receptive field and the surround, indicating that remarkably similar principles drive attention modulation inside the center and surrounding regions of the receptive field. A spatially-tuned normalization model, fitted to the responses of individual neurons, captured the dependency of attention modulation on both stimulus selectivity and suppression, and provided an excellent account of how attention operates across different regions of the receptive field, with either single or multiple stimuli shown inside. Each stimulus configuration induced variable amounts of excitation and suppression in different neurons, depending on the receptive field position of the stimuli. Attention operates on this variable excitation and suppression, thereby explaining why the magnitude of attention-related modulations varies so widely across neurons.

*Reynolds et al. (1999)* observed a strong correlation between stimulus selectivity and their index of stimulus-induced suppression. It is important to note that this strong correlation is not a general property of spatial summation in visual cortex. Instead, the correlation they observed likely arose from the experimental design used in that study. Specifically, for each neuron Reynolds and colleagues presented different stimulus pairs that fell on two locations that were always chosen to lie well within the cRF at similar distances from the receptive field center. The authors kept the stimulus at one location fixed (reference stimulus), but varied the orientation and color of the stimulus at the second location. But varying the orientation and color of a stimulus, but not its location, varies predominantly the excitation and not the suppression to the neuron (see Appendix; we found no evidence for orientation-tuned suppression in our data). Consequently, both the selectivity and the stimulus-induced suppression index varied with a single variable, the strength of the second stimulus relative to the strength of the fixed stimulus. This explains the strong correlation between stimulus selectivity and suppression in that study. In contrast, we presented stimuli at locations across both the classical and surrounding receptive field, allowing both stimulus-induced excitation and suppression to vary across stimulus configurations. This is the primary reason for the difference between the studies. The variable reference stimulus across neurons in Reynolds et al., and the slightly different indices used in the two studies will have further amplified the differences in findings in both studies. Hence, under more general stimulus conditions wherein stimuli can fall on any receptive field location, stimulus selectivity and stimulus-induced suppression are not correlated.

Importantly, by avoiding a correlation between selectivity and suppression, we were able to examine their separate contributions to the magnitude of attention modulation. Shifting attention between a strong and a weak stimulus, each presented at different receptive-field locations, changes which neuronal inputs are emphasized, thereby causing neuronal-response modulations. However, we show that such clear-cut differential processing only occurs if the weaker stimulus also induces strong suppression: without suppression, attention has no leverage to amplify input differences. The spatially-tuned normalization model captures the dependency of attention-related modulation on both selectivity and suppression, and does so for neurons with stimuli in either the cRF or the surround.

Both *Sundberg et al. (2009)* and *Sanayei et al. (2015)* used conditions with one stimulus inside the classical receptive field and at least one other stimulus in the surround (sRF-cRF). However, neither of these studies used a condition with both stimuli positioned inside the cRF (cRF-cRF). Hence, a direct comparison of attention modulation within the cRF and the surround could not be performed in these studies. This comparison is crucial to determine whether the neuronal effects of attention differ between the cRF and the surround. Importantly, we found that seeing the similarity between attentional effects in both receptive field regions requires examination of the combined relationship, i.e. interaction, between attention modulation and both stimulus selectivity and stimulus-induced suppression (*Figure 4 versus 5*). The interaction between these variables was similar in both receptive field configurations.

Sanayei et al. fit different (normalization) models, but never measured the neuronal effects of attention when attention was directed to the surround stimulus; the authors only compared the effects of attention to the cRF stimulus, with or without surround stimulation, versus attention to a distant stimulus. Attention was never directed to the surround stimuli. Thus, Sanayei et al. lacked the crucial information needed to examine how surround suppression affects attention modulation and to test the efficacy of normalization models. We tested whether a single model could fit both the cRF-cRF and sRF-cRF data. The good fits of the spatially-tuned normalization model to the data obtained in both receptive field configurations provided further evidence that attention acts similarly inside the cRF and the surround.

Suppression and excitation may rely on distinct mechanisms in different regions of the receptive field (*Angelucci et al., 2014*). Our data do not pertain to these different mechanisms and we may have missed some small differences in attention modulation associated with these distinct mechanisms. Nonetheless, our findings show that the way attention interacts with excitation and suppression across different regions of the receptive field is remarkably similar.

In recent years, several normalization models of attention have been proposed (*Reynolds et al., 1999*; *Ghose and Maunsell, 2008*; *Ni et al., 2012*; *Ghose, 2009*; *Reynolds and Heeger, 2009*; *Lee, 2009*; *Boynton, 2009*; *Lee et al., 1999*). Two of the more elaborate models explicitly assumed that attention acts on a specific receptive-field structure, namely one that encompasses a relatively

narrow excitatory field in addition to a wider suppressive field (*Ghose, 2009*; *Reynolds and Heeger, 2009*). This receptive-field structure is based on findings from primary visual cortex (V1) (*Cavanaugh et al., 2002b*; *Sceniak et al., 1999*; *DeAngelis et al., 1994*). It is important to note, however, that none of these studies empirically tested if attention actually operates on the spatially-varying excitation and suppression implied by such a receptive field structure. We started with a spatially-tuned normalization model that made no assumptions about the structure of excitation and suppression in the receptive field of V4 neurons. Furthermore, and in contrast to the earlier-mentioned studies (*Ghose, 2009*; *Reynolds and Heeger, 2009*), we explicitly fitted models to the responses of individual neurons to test relationships with the underlying receptive field structure. Interestingly, this naive model reveals that the receptive field organization of V4 neurons strongly resembles that of V1 neurons (*Cavanaugh et al., 2002b*; *Sceniak et al., 1999*; *DeAngelis et al., 1994*): both excitation and suppression are maximal near the receptive-field center, but excitation is more spatially concentrated, while suppression stretches over larger distances. These findings suggest that similar receptive field organizations can be found throughout different stages of visual cortex. Importantly, our findings show that attention operates uniformly across the spatially-varying excitation and suppression of a receptive field: throughout the receptive field, including the surround, attention-related modulations of neuronal responses is governed by very similar normalization rules.

The finding that the rules of neuronal attention modulation are similar across different regions of the receptive field simplifies our view of attentional operations in visual cortex, and provides strong support for normalization models of attention (*Ghose, 2009*; *Reynolds and Heeger, 2009*). We also show that the origin of the stimulus-induced excitation is not important for determining the magnitude of attention modulation: we found no distinction between excitation related to a neuron's feature tuning (i.e. orientation tuning) or spatial tuning (i.e. receptive field). What matters for neuronal attention modulation is stimulus-induced excitation, regardless of its origin, in conjunction with spatially-tuned suppression. It follows that when a particular stimulus configuration induces variable amounts of excitation and suppression in different neurons, attention-related modulations will vary across these neurons.

The fact that attention operates on the spatially-varying excitation and suppression of a receptive field has important implications, as it determines which neurons will be most influenced by attention. For instance, with a given number of stimuli presented inside the receptive field, attention to a preferred stimulus shown inside the center of the receptive field typically has the greatest potential to elevate neuronal responses. This is true not only because stimuli near the receptive field center generally elicit most excitation, but also because such stimuli are most likely to induce the greatest suppression. The elevated suppression by center stimuli gives them more weight in normalization mechanisms as it allows them to better discount the suppressive influences from other simultaneously presented stimuli. Similarly, attention to a weak stimulus inside the receptive field center will in general reduce responses more than attention to a weak stimulus elsewhere in the receptive field, including the surround. This does not mean that stimuli in the surround, which induce relatively less suppression, have little impact on attention modulation. Indeed, because the surround is so much larger than the cRF it can contribute considerable suppression. Such strong surround suppression likely occurs under natural viewing conditions where stimuli are shown throughout the visual field, many of them covering the surround (*Vinje and Gallant, 2000*; *Ozeki et al., 2009*; *Haider et al., 2010*; *Coen-Cagli et al., 2015*). The normalization model predicts that such strong surround suppression may robustly amplify attention modulation, much beyond the attention modulation observed without surround suppression.

This effect is illustrated in *Figure 8* in which spatial attention was applied to model neurons *with* (upper panels *Figure 8*) or *without* a surround (lower panels *Figure 8*). The model neurons with a surround strongly modulated their responses by attention, but those without a surround much less (*Figure 8F* upper vs. lower panel). Hence, although the precise role of the surround is still unknown (*Schwartz and Simoncelli, 2001*; *Vinje and Gallant, 2000*; *Sachdev et al., 2012*), an important contribution of the surround may lie in its ability to amplify attention-related response modulations.

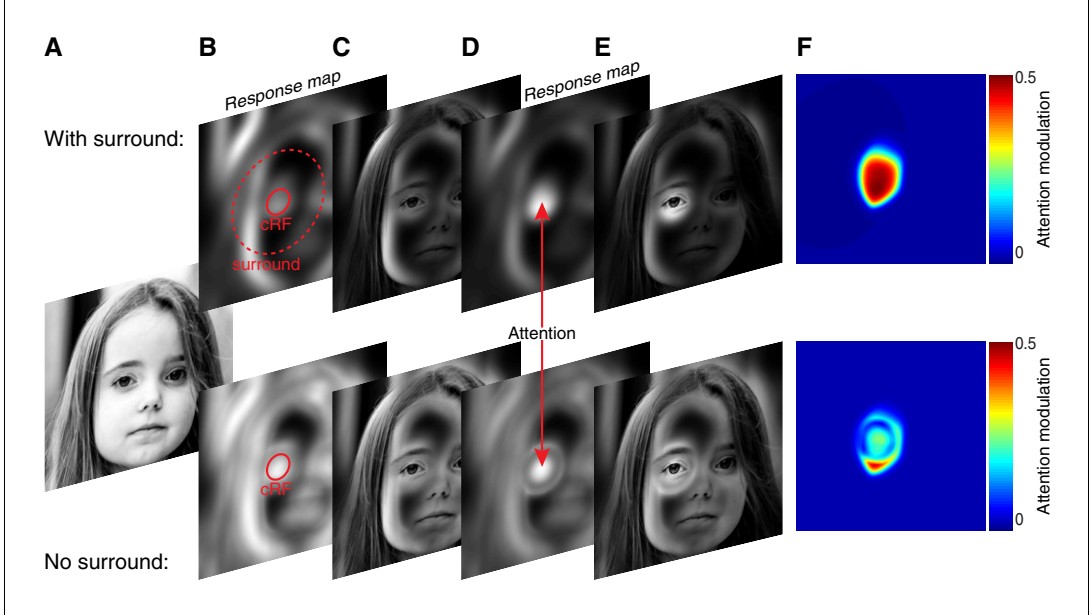

**Figure 8.** The surround may amplify spatial attention under natural viewing conditions. (**A**) Original image. (**B**) Model neurons tiled the image. Each pixel contained one model neuron with its receptive-field centered on that pixel. An example cRFs (solid red) and surround (red dashed) for one neuron are shown. The radius of each neuron's surround was approximately five times larger than the radius of its cRF. The model neurons computed local contrast within the excitatory and suppressive component of their receptive field. The response maps show each neuron's response: neurons near high-contrast regions responded most as indicated by the luminance of the pixels. (**C**) Original image scaled according to the response map in (**B**). (**D**) Attention was directed to the left eye. Attention weighed the excitatory and suppressive inputs with its Gaussian kernel, resulting in stronger responses of the neurons with receptive fields near the attended location relative to neurons with receptive fields outside the locus of attention. (**E**) Original image scaled according to the response map in (**D**), illustrating the way attention changes the visual representation. (**F**) Attention modulation of each neuron, defined as (response$^{Att}$ - response) / (response$^{Att}$ + response). Here, *response* is the response map without attention as in (**B**), while *response*$^{Att}$ is the response map with attention as in (**D**). Upper panels (**B–F**) are based on model neurons with a suppressive surround. Lower panels (**B–F**) are based on model neurons without a suppressive surround, but with the same amount of suppression inside the cRF as the neurons with a suppressive surround.

## Materials and methods

### Surgical procedures

Two male rhesus monkeys M1 and M2 (*Macaca mulatta*, both 9 kg) were trained to perform a spatial attention task. Monkeys were pair housed in standard 12:12 light-dark cycle and given food *ad libitum.* Before training, each animal was implanted with a head post. After completion of the behavioral training (~7 months), we implanted a 10 × 10 array of microelectrodes into area V4 of the left cerebral hemisphere, between the lunate sulcus and the superior temporal sulcus. Before surgery, animals were given buprenorphine (0.005 mg/kg, intramuscular) and flunixin (1.0 mg/kg, intramuscular) as analgesics, and a prophylactic dose of an antibiotic (Baytril, 5 mg/kg, intramuscular). For surgery, animals were sedated with ketamine (15 mg/kg, intramuscular) and xylazine (2 mg/kg, intramuscular) and given atropine (0.05 mg/kg, intramuscular) to reduce salivation. Anesthesia was maintained with 1–2% isoflurane. Antibiotic was administered again 1.5 hr into surgery; buprenorphine and flunixin were given for 48 hr post-operatively. All procedures were approved by the Institutional Animal Care and Use Committee of Harvard Medical School (Boston, MA; protocol #04214).

### Visual stimulation

Stimuli were presented on a gamma-corrected cathode-ray tube (CRT) display with a 100 Hz frame rate and a resolution of 1024 × 768 pixels. Monkeys were seated 57 cm from the center of the screen. Stimuli consisted of full-contrast achromatic odd-symmetric static Gabor stimuli (0.6–2.2 cycles per degree; one spatial frequency per daily session) presented on a gray background (42 cd/m [*Kastner and Ungerleider, 2000*]) and were rendered online using custom-written software

(https://github.com/MaunsellLab/Lablib-Public-05-July-2016.git). The Gabor stimuli were truncated at three SD from their center.

## Spatial attention task

We trained monkeys to perform a visual detection task in which spatial attention was manipulated (*Figure 1A*). The trial started when the monkey fixated a small spot in a virtual 1.5° square fixation window in the center of the video display for 240–700 ms. Eye movements were tracked using an infrared eye-tracking camera (EyeLink 1000) sampling binocularly at 500 Hz. The duration of the fixation period was randomly drawn from a uniform distribution. Following fixation a sequence of stimuli was presented, in which each stimulus presentation lasted 200 ms and was separated from other stimuli by 200–1020 ms interstimulus intervals (*Figure 1B*). The durations of the interstimulus intervals were randomly drawn from an exponential distribution ($\tau$ = 200 ms). During the interstimulus interval only a gray screen with the fixation dot was shown. The stimulus presentations were short to prevent animals from adjusting their attention within a stimulus presentation in response to the number of stimuli presented (*Lee and Maunsell, 2010*; *Ni et al., 2012*; *Lee, 2009*; *Williford and Maunsell, 2006*).

On each trial, stimuli appeared at two locations near the receptive fields of neurons, but the two locations differed between blocks of trials. One stimulus location (the middle location: location 1 in *Figure 2A*) never varied, but in different blocks of trials the second stimulus location was shifted either clockwise (location 2 in *Figure 2A*) or counterclockwise (location 3 in *Figure 2A*). For the example session in *Figure 2A*, the possible stimulus-location pairings were 1+2 and 1+3. All stimulus locations were equidistant from the fixation point, and stimulus locations 2 and 3 were equidistant from stimulus location 1. The two different pairs of stimulus locations assured that many neurons were tested in both receptive-field configurations (cRF-cRF and sRF-cRF).

On each stimulus presentation within a trial, we presented one, two, or no stimuli at the two stimulus locations near the neurons' receptive fields. The stimuli could be of one of two orthogonal orientations. Each session, the stimulus orientation was optimized for a randomly selected unit, so that different orientations were used across sessions. A representative set of nine possible stimulus combinations (for a particular orientation pair) is shown in *Figure 2B*. Using these different stimulus combinations we could measure stimulus selectivity, stimulus-induced suppression and attention modulation.

Each stimulus location near the neurons' receptive fields (stimulus location 1, 2, 3 in *Figure 2A*) had a corresponding and equally eccentric stimulus location on the opposite side of the fixation point (e.g. stimuli near *Away* in *Figure 2C,D*). As outlined below, we instructed monkeys to direct their attention to one stimulus location, either near or away from the receptive fields. This way we could measure not only how attention modulated neuronal responses when directed to different stimuli near the neurons' receptive fields, but also measure stimulus selectivity and stimulus-induced suppression with attention directed away from the neurons' receptive fields.

On each stimulus presentation (of multiple in a trial), each stimulus location was equally likely to contain one orientation, the other orientation or no stimulus. When Gabor pairs were presented near the neurons' receptive fields, their centers were separated by a median of 2.3° (range: 1.6–4.8°), and always separated by at least six Gabor standard deviations (mean Gabor $\sigma$: 0.45°; range: 0.17–0.5°). With such inter-stimulus distances, two stimuli can be presented within the receptive fields of V4 neurons.

Subjects were required to detect a faint white spot, labeled *Target* in *Figure 1A,B*. The target appeared at one of the four stimulus locations (two near the neurons' receptive field and two counterparts on the opposite side of the fixation point; see above) during one stimulus presentation within a trial. The target never appeared on the first stimulus presentation of a trial, but could occur with equal probability on any other stimulus presentation (range: 2–8). Two to five percent of the trials contained no target and the monkey was rewarded for maintaining fixation. Targets were presented in the center of Gabor stimuli to encourage the monkeys to confine their attention to a restricted part of visual space, near the cued stimulus location.

Task difficulty was manipulated by varying the target strength, defined as the opacity of the target (range of alpha-transparency values: 0.06–0.28). Each session we used six different target strengths (*Figure 1B,C*). The monkey was rewarded with a drop of juice for making a saccade to the target location within 350 ms of its appearance.

Attention was cued to one location in blocks of ~150 trials. Before the start of each block the monkey performed three to five instruction trials in which stimuli were presented at a single (cued) location. The instruction trials cued the monkey to attend to that location during subsequent trials in which stimuli could occur at all four locations.

Within a block of trials, the target appeared at the cued location in 91% of the trials (valid trials; position of the black circle in *Figure 1A*). In the remaining 9% of the trials (invalid trials) the target appeared at one of the three other (uncued) stimulus locations, with equal probability (position of the yellow and blue circles in *Figure 1A*). We used a single target strength for the invalid trials, as this allowed us to obtain reliable estimates of behavior at the unattended locations despite the small number of invalid trials (*Figure 1B,C*) (*Cohen and Maunsell, 2009*). Using invalid trials, we could compare performance between attended and unattended locations.

## Recordings

We recorded neuronal activity using a 10 × 10 array of microelectrodes (Blackrock Microsystems; impedances: 0.3–1.2 MΩ at 1 kHz; 1 mm long electrodes; 0.4 mm between adjacent electrodes), chronically implanted into area V4 of the left cerebral hemisphere of each monkey. The data presented here are from 130 daily sessions of recording (Monkey M1: 52; Monkey M2: 78).

At the beginning of each recording session, we mapped the receptive fields and optimized stimulus parameters (position, orientation) for a randomly selected unit. We first measured the orientation-tuning curve of each neuron using a large Gabor that covered the lower right visual field. Orientation tuning was measured using Gabors of 8 different orientations spanning 180°.

We then mapped the spatial receptive field of each neuron using a Gabor with the preferred orientation of the selected unit, and a Gabor with an orientation orthogonal to that neuron's preferred orientation. Using two orthogonal orientations assured that most neurons were responsive to at least one stimulus. The same two orientations were subsequently used during the attention task. Receptive-field mapping was computer controlled and used a full-contrast static Gabor with the two orthogonal orientations (spatial frequency: 1.1 cycles per degree; Gabor sigma: 0.3°) on an 8 by 8 grid of positions (~azimuth range: −1 to 8°; ~elevation range: 2.5 to −8°). The center-of-mass of the receptive field (unfitted data) was defined as the receptive-field center. The receptive field plots in *Figure 3* are based on the linearly-interpolated spike counts measured at each grid location. The spike counts were obtained within a 200 ms window starting 50 ms after stimulus onset.

Action potential waveforms were sorted offline using spike-sorting software (Offline Sorter, Plexon Inc). Waveforms for which the first two principal-component scores formed a well-defined cluster, separate from other waveforms, were classified as single units. The receptive fields of the units were located in the lower right quadrant at an average eccentricity of 3° for monkey M1 and 4° for monkey M2.

## Analyses

We included only neuronal data from stimulus presentations from correct, validly-cued trials. We excluded incorrect trials, invalidly-cued trials, instruction trials, trials with no target, the first stimulus presentation of a trial (on which no target could occur), and stimulus presentations with a target. Neurons were included in the analyses if they responded significantly above baseline to any single Gabor presented at any stimulus location in the *attend away* condition (ANOVA; α=0.05). Responses in the *attend away* condition were obtained by averaging the firing rates from the conditions in which attention was directed to either of the two stimulus locations furthest away from the receptive-field center of the neuron (*Away* in *Figure 2C,D*). The small subset of neurons whose responses where significantly suppressed below baseline by all stimuli (N=13) was not further analyzed. Neuronal responses were computed based on the spikes in the interval from 50 ms to 300 ms after stimulus onset. Similar results were obtained using different intervals.

A stimulus location was considered within the classical receptive field (cRF) if the neuron responded significantly to any single stimulus (of either orientation) presented at that location, measured with attention away from the neuron's receptive field (*attend away*). The median distance from the receptive-field center of a stimulus inside the cRF was 1.7° (interquartile range 1.2° to 2.5° or 0.7–1.5σ, where σ is the Mahalanobis distance from those neurons whose receptive fields were well fitted with a bivariate Gaussian function: >80% explained variance, N=306 neurons). A stimulus

location was considered to be within the surround of a neuron if the neuron did not increase its firing rate significantly to any single stimulus (of either orientation; N>36 trials per stimulus) presented at that location, measured with attention away from the neuron's receptive field (*attend away*). Note that neurons for which a surround location was measured did respond significantly to at least one of the stimuli when it was presented inside the cRF instead of the surround (*Figure 2—figure supplement 1*). The median distance of a surround location from the receptive-field center was 3.5° (interquartile range 2.7° to 4.3°, or 1.9σ to 3.1σ). We obtained similar results when we additionally required surround positions to lie at more than 2.5σ from the receptive field center. We also recorded from neurons with two stimuli inside their surround, i.e. sRF-sRF configuration. These data were not further analyzed due to a lack of responses.

The peristimulus time histograms (PSTHs) in *Figure 3A–D* were computed by counting the number of spikes in 1 ms bins and smoothing with a Gaussian filter of σ = 5 ms.

A selectivity index was computed based on the responses to the component Gabors of each Gabor pair (four pairs in *Figure 2B*). Selectivity indices were computed for each Gabor pair presented at each pair of stimulus locations (stimulus locations 1+2 or 1+3 in *Figure 2A*). We thus obtained eight selectivity indices per neuron. The selectivity index is defined as (P - N)/(P + N). Here, P (preferred) and N (non-preferred) are the neuronal responses to the strongest and weakest component Gabor of a Gabor pair when presented alone. P and N were measured with attention away from the neurons' receptive fields (*attend away*). The upper pictogram in *Figure 2E* illustrates the computation of the selectivity index for one Gabor pair. By definition the neuron does not respond to the stimulus when it appears alone inside the surround. It follows that for the sRF-cRF receptive-field configuration, the surround Gabor is always assigned as non-preferred (N) and the cRF Gabor as preferred (P).

Stimulus-suppression indices were similarly computed for each of eight possible Gabor pairs as (P - PN) / (P + PN), where P is the response to the preferred Gabor of a Gabor pair as described before, and PN is the response to the Gabor pair. Both P and PN were measured with attention away from the neurons' receptive fields (*attend away*). The middle pictogram in *Figure 2E* illustrates the computation of the stimulus-induced suppression index for one Gabor pair. We obtained similar results when defining a suppression index as (P+N-PN) / (P+N+PN). Note that the stimulus-induced suppression index is distinct from the $\alpha$ terms in the model. This is because the stimulus-induced suppression index is based on the observed neuronal responses, which comprise both an $\alpha$ and $L$ term (i.e. response = $L/(\alpha+\sigma)$). In terms of the model parameters, the stimulus-induced suppression index is given by:

$$Stimulus-induced\ suppression\ index = \frac{response_P - response_{P+N}}{response_P + response_{P+N}} = \frac{\frac{L_P}{\alpha_P+\sigma} - \frac{L_P+L_N}{\alpha_P+\alpha_N+\sigma}}{\frac{L_P}{\alpha_P+\sigma} + \frac{L_P+L_N}{\alpha_P+\alpha_N+\sigma}},$$

where $L_P$ is the excitatory drive from the preferred component Gabor of a Gabor pair, $L_N$ is the excitatory drive from the non-preferred component Gabor, $\alpha_P$ is the suppressive drive from the preferred component Gabor, and $\alpha_N$ is the suppressive drive from the non-preferred component Gabor. So the stimulus-induced suppression index depends on both the excitatory and suppressive drive from the stimulus.

Attention-modulation indices were computed for each of eight possible Gabor pairs as ($P^{Att}N$ - $PN^{Att}$) / ($P^{Att}N$ + $PN^{Att}$), where $P^{Att}N$ is the neuronal response to the Gabor pair with attention directed to P, $PN^{Att}$ is the neuronal response to the Gabor pair with attention directed to N. The lower pictogram in *Figure 2E* illustrates the computation of the attention-modulation index for one Gabor pair.

All Gabor pairs for which a neuron responded on average with at least 1 spike (in the 250 ms window) in the *attend away* condition were further analyzed, but similar results were obtained using other criteria. This way neuronal data from 728 neurons were analyzed (monkey M1: 264; M2: 464). In *Figures 4*, *5* selectivity and stimulus-induced suppression indices are computed for each neuron and all Gabor pairs, so neurons contribute more than one index. Due to the chronic nature of our recordings, it is likely that some neurons were resampled across days. Because we adjusted the values of the stimulus orientations and locations each day for a randomly selected unit, resampling rarely involved identical stimulus configurations. Similar results were obtained for both monkeys (see Results).

We used multiple linear regression to examine if attention-related modulation depends on the interaction between stimulus selectivity and stimulus-induced suppression. For both RF configurations, the model included a main effect of selectivity, supplemented with a main effect of stimulus-induced suppression. The model also included an interactive product term, which measured the dependency of attention-related modulation on both selectivity and stimulus-induced suppression. The regression model is given by:

$$\text{attention modulation} = \text{selectivity} \cdot \beta_1 + \text{suppression} \cdot \beta_2 + \text{selectivity} \cdot \text{suppression} \cdot \beta_3 + error$$

In this model, the main effect of selectivity measures the contribution of selectivity to attention modulation given that suppression is zero. Similarly, the main effect of suppression measures the contribution of suppression to attention modulation given that selectivity is zero. For example, if suppression is zero, the suppression and the interaction term (selectivity $\times$ suppression $\times$ $\beta_3$) both go to zero, leaving only the selectivity term $\beta_1$, which specifies the contribution of selectivity to attention modulation. Conversely, if selectivity is zero, the only non-zero term is the $\beta_2$ term, which specifies the contribution of suppression to attention modulation. Thus the main effects are not estimated from a particular selection or a subset of the dataset, but follow mathematically from the linear regression model with interaction term. For the Bayesian regression analysis, we compared the marginal likelihood of the data given a regression model that does not include receptive field configuration as a factor to the marginal likelihood of the data given a regression model that does include receptive field configuration as a factor (i.e. the Bayes factor, using the lmBF function from the BayesFactor package in R [*Rouder et al., 2012*]).

The plots in *Figure 5A,B,D,E* were obtained using regularized bilinear interpolation on the observed or modeled attention-modulation indices from all Gabor pairs and all neurons (Surface Fitting using gridfit (http://www.mathworks.com/matlabcentral/fileexchange/8998), MATLAB Central File Exchange).

## Model

Tuned normalization has been applied before (*Carandini et al., 1997*; *Schwartz and Simoncelli, 2001*; *Lee et al., 1999*; *Rust et al., 2006*) and has been used to explain neuronal-response modulation when attention is shifted between two stimuli with different motion directions inside the cRF of MT neurons (*Ni et al., 2012*). The spatially-tuned normalization model is described by *Equation (1)*. This spatially-tuned normalization model was fitted to the neuronal responses of all 728 neurons used in the analyses. The model parameters are: $L_{11}$, $L_{12}$, $L_{21}$, $L_{22}$, $L_{31}$, $L_{32}$, $\alpha_2$, $\alpha_3$, $\sigma$, $\beta$. Specifically, $L_{ij}$ (adopted from *linear* response) is the excitatory drive from a stimulus of orientation $j$ ($j$ = 1, 2) at receptive-field location $i$ ($i$ = 1, 2, 3). $\alpha_p$ is the suppression parameter associated with stimulated receptive-field location p=1, 2, 3). $\beta$ adds attention to the model and is multiplied with the parameters associated with the attended location ($L$ and $\alpha$; see Results and *Figure 5C*). In the conditions with attention directed away from the receptive fields (*attend away*) $\beta$ = 1 (see *Equation (1)*). $\sigma$ is the semi-saturation constant that is fixed across conditions and serves as a baseline suppression parameter. The $\sigma$ parameter was introduced in Heeger's (*Heeger, 1992*) original divisive normalization model to model the shape of contrast-response functions of neurons in primary visual cortex. It also stabilized the response when stimuli of low (or zero) contrast are presented by preventing division by zero. For our data, which involve only high-contrast stimuli, it represents baseline suppression, which may arise from spontaneously active inhibitory neurons or suppression caused by constant stimuli (e.g. the edge of the stimulus display) visible to the monkey. It is an important parameter to accommodate the neuronal effects of attention to single stimuli inside the receptive field (see *Figure 5—figure supplement 1A vs. B*: P vs. P$^{att}$). The median value of the sigma parameter was 0.06 (median absolute deviation (MAD): 0.26). A model with no sigma term performs significantly worse at explaining responses to isolated attended stimuli (median two-fold cross-validated percentage explained variance 84%, compared to 87% for the model with sigma term; p=0.006; sequential F-test). We also fit for each neuron a model with only one free excitatory ($L$) term capturing excitation across all stimulus conditions. This model with one $L$ term performed significantly worse at explaining neuronal responses than the full model with all $L$-terms (median two-fold cross-validated percentage explained variance 48%, compared to 87% for the model with all $L$ terms; p<0.0001; sequential F-test).

We tested two pairs of receptive-field locations (see Spatial attention task and *Figure 2A*). $\alpha_1$ is the suppression parameter related to the receptive-field location common to both of the receptive-field location pairs (stimulus location 1 for the example session in *Figure 2A*), and is set to one to constrain the model. All parameters were constrained to be nonnegative. For each neuron, the model was fitted to the neuronal responses of 36 distinct attention and stimulus combinations by minimizing the sum of squared error using a simplex optimization algorithm (MATLAB fminsearch.m; MathWorks). The goodness-of-fit of the model was calculated for each neuron as the percentage explained variance, which was determined by taking the square of the correlation coefficient between the model-predicted responses and the observed neuronal responses across all stimulus conditions. The explained variance was calculated using the average neuronal responses from trials not used to fit the model. For this purpose, we employed two-fold cross-validation, fitting the model based on half of the randomly-chosen trials, and using the remaining data to measure the goodness of fit of the model. This procedure was repeated five times, each time using a different random draw, and subsequently averaged across all cross-validations to produce the reported goodness-of-fit.

In *Figure 8*, each image pixel contained one model neuron with its receptive field centered on that pixel. The neurons' receptive fields consisted of an excitatory and suppressive receptive field. These receptive fields were modeled as a circular two-dimensional Gaussian with a standard deviation of eight pixels for the excitatory field and 40 pixels for the suppressive field. The model neurons computed local contrast within their excitatory and suppressive receptive field. The excitatory input was divisively normalized by the suppressive input to generate the model neuron's response. Model neurons without a surround experienced no suppression from stimuli positioned outside their classical receptive field. The classical receptive field was defined as all pixels within 16 pixels, i.e. two standard deviations from the excitatory receptive field, from the receptive-field center. Attention was modeled as a circular two-dimensional Gaussian kernel with a standard deviation of five pixels and amplitude equal to six. Attention weighed the excitatory and suppressive inputs with its kernel, resulting in stronger responses of the model neurons near the locus of attention relative to model neurons outside the locus of attention.

## Acknowledgements

We thank Thomas Zhihao Luo for good discussions and comments on the manuscript. We thank Jackson J Cone, Till S Hartmann, Mark H Histed, J Patrick Mayo and Amy M Ni for comments on an earlier version of the manuscript, and Steven J Sleboda for technical assistance. Bram-Ernst Verhoef is a postdoctoral research fellow of the Flemish Fund for Scientific Research (FWO).

## Additional information

### Funding

| Funder | Grant reference number | Author |
| --- | --- | --- |
| Fonds Wetenschappelijk Onderzoek | | Bram-Ernst Verhoef |
| National Institutes of Health | R01EY005911 | John HR Maunsell |
| National Institutes of Health | R01EY021550 | John HR Maunsell |

The funders had no role in study design, data collection and interpretation, or the decision to submit the work for publication.

### Author contributions

B-EV, Conception and design, Acquisition of data, Analysis and interpretation of data, Drafting or revising the article; JHRM, Conception and design, Analysis and interpretation of data, Drafting or revising the article

## Author ORCIDs

Bram-Ernst Verhoef, http://orcid.org/0000-0003-3535-9008

John HR Maunsell, http://orcid.org/0000-0003-0018-4439

## Ethics

Animal experimentation: This study was performed in strict accordance with the recommendations in the Guide for the Care and Use of Laboratory Animals of the National Institutes of Health. All procedures were approved by the Institutional Animal Care and Use Committee of Harvard Medical School (Boston, MA; protocol #04214).

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

# Appendix

## Selectivity and suppression are correlated when only one of two stimuli is varied

*Equation 1* from the spatially-tuned normalization dictates that suppression and selectivity will be correlated when only one of two stimuli is varied. In the spatially-tuned normalization model, the $L$ and $\alpha$ terms are independent of each other. However the $\alpha$ terms are fixed at each RF location. Consequently, varying only the stimulus at one RF location, while keeping the other stimulus fixed, corresponds to keeping $L_1$, $\alpha_1$ and $\alpha_2$ fixed. Here, $L_1$ is the excitatory drive from the fixed stimulus, $\alpha_1$ is the suppressive drive from the fixed stimulus, and $\alpha_2$ is the suppressive drive from stimulating the RF location from the variable stimulus. Thus $L_2$, the excitatory drive from the variable stimulus, is the only variable that varies when different stimuli are presented at that RF location. The variable $L_2$ term changes both the selectivity and suppression indices in similar ways. In model terms, this can be seen as:

$$selectivity = \frac{response_1 - response_2}{response_1 + response_2} = \frac{\frac{L_1}{\alpha_1+\sigma} - \frac{L_2}{\alpha_2+\sigma}}{\frac{L_1}{\alpha_1+\sigma} + \frac{L_2}{\alpha_2+\sigma}},$$

where we assume (without loss of generality) that the fixed stimulus at location 1 is the preferred stimulus, i.e. $\frac{L_1}{\alpha_1+\sigma} > \frac{L_2}{\alpha_2+\sigma}$. Also:

$$Stimulus - induced\ suppression = \frac{response_1 - response_{1+2}}{response_1 + response_{1+2}} = \frac{\frac{L_1}{\alpha_1+\sigma} - \frac{L_1+L_2}{\alpha_1+\alpha_2+\sigma}}{\frac{L_1}{\alpha_1+\sigma} + \frac{L_1+L_2}{\alpha_1+\alpha_2+\sigma}},$$

where the fixed stimulus at location 1 is again the preferred stimulus.

If we present a weak stimulus at the variable location 2, i.e. $L_2 \approx 0$, the selectivity and stimulus-induced suppression index both become high:

$$selectivity = \frac{response_1 - 0}{response_1 + 0} = \frac{\frac{L_1}{\alpha_1+\sigma} - \frac{0}{\alpha_2+\sigma}}{\frac{L_1}{\alpha_1+\sigma} + \frac{0}{\alpha_2+\sigma}} = \frac{\frac{L_1}{\alpha_1+\sigma}}{\frac{L_1}{\alpha_1+\sigma}} = 1,$$

and

$$Stimulus - induced\ suppression = \frac{response_1 - response_{1+2}}{response_1 + response_{1+2}} = \frac{\frac{L_1}{\alpha_1+\sigma} - \frac{L_1+0}{\alpha_1+\alpha_2+\sigma}}{\frac{L_1}{\alpha_1+\sigma} + \frac{L_1+0}{\alpha_1+\alpha_2+\sigma}}$$
$$= \frac{\frac{L_1}{\alpha_1+\sigma} - \frac{L_1}{\alpha_1+\alpha_2+\sigma}}{\frac{L_1}{\alpha_1+\sigma} + \frac{L_1}{\alpha_1+\alpha_2+\sigma}}$$

The stimulus-induced suppression index is also high because $response_{1+2} = \frac{L_1+L_2}{\alpha_1+\alpha_2+\sigma}$ is at its lowest (less subtraction in the numerator) when stimulus 2 adds no excitatory drive to $response_{1+2}$.

Conversely, adding a more potent stimulus at variable location 2, i.e. increasing $L_2$, decreases the selectivity, because $response_2$ is now not zero anymore and thus decreases the numerator. Similarly, the stimulus-induced suppression index will also be smaller, because $response_{1+2} = \frac{L_1+L_2}{\alpha_1+\alpha_2+\sigma}$ increases with increasing $L_2$, which in turn decreases the numerator, $\frac{L_1}{\alpha_1+\sigma} - \frac{L_1+L_2}{\alpha_1+\alpha_2+\sigma}$, of the stimulus-induced suppression index.

Thus, both selectivity and stimulus-induced suppression increase with a weaker stimulus at variable position 2, and both selectivity and stimulus-induced suppression decrease with a

more potent stimulus at variable position 2. This shows that, within cells, selectivity and stimulus-induced suppression will be correlated when only one of two stimuli is varied. This agrees with the single neuron examples in Reynolds et al.

Reynolds et al. repeated these measurements for several neurons and included all data points from each neuron into the populations scatter plots. Note that Reynolds et al. always positioned their stimuli at approximately equally-responsive RF locations. The RFs of V4 neurons are approximately Gaussian, which means that equally-responsive RF positions are expected to lie at approximately equal (Mahalanobis) distances from the RF center. Following *Figure 7*, these equally-responsive RF positions, at equally distant RF positions, are expected to have similar alphas. Because Reynolds et al. only sampled RF locations with similar alphas, similar positive relationships between selectivity and suppression would have been observed for each neuron. Given that all neurons had a similar positive relationship between selectivity and suppression, the population data will also display a similar positive relationship.

Our experimental design differed substantially from that of Reynolds et al., because when one stimulus was fixed there were few variants of the other stimulus. For example, when presenting a Gabor with one orientation at RF location 1, our design introduced only two different stimuli at RF location 2: a Gabor with the same orientation or a Gabor with an orthogonal orientation as that at location 1. In contrast, Reynolds et al. used 16 different stimulus conditions.

We compared the two stimulus conditions in which the stimulus at one location was kept constant while the stimulus at the other location varied between two orthogonal orientations. Across neurons and stimuli, we found that the average stimulus-induced suppression index of the stimulus condition with greater selectivity was significantly greater than that of the other stimulus condition with the lower selectivity (stimulus-induced suppression index of 0.07 vs. 0.05, $p = 3 \times 10^{-13}$, paired permutation t-test). Thus, despite the very limited number of stimulus conditions in our design, we could reproduce the correlation between suppression and selectivity of that paper.

