## [Decision Letter]

[Editors’ note: a previous version of this study was rejected after peer review, but the authors submitted for reconsideration. The first decision letter after peer review is shown below.]

Thank you for submitting your work entitled "Attention operates uniformly throughout the Classical Receptive Field and the Surround" for consideration by *eLife*. Your article has been reviewed by three peer reviewers, and the evaluation has been overseen by a Reviewing Editor and David Van Essen as the Senior Editor. Our decision has been reached after consultation between the reviewers. Based on these discussions and the individual reviews below, we regret to inform you that your work will not be considered further for publication in *eLife* in its present form due to significant concerns about the validity of the central claims, as detailed in the reviews. However, all three reviewers believe that the work has high potential to make a significant contribution, therefore we invite you to submit a new version of the manuscript if you are able to address the central comments of the reviewers.

Reviewer #1:

Summary:

I found this study to be very elegant in its linking of several complex phenomena (selectivity, suppression, and attentional modulation) with a single, simple model. The paper is also beautifully written. Verhoef and Maunsell performed array recordings in macaque to study the effects of attention on stimuli either inside the classical receptive field of a neuron or in the surround. They show that attention acts uniformly on the response of neurons by amplifying the effect of the attended stimulus, i.e. increasing the response for the preferred stimulus and exacerbating the suppression for a non-preferred stimulus, regardless of whether it is in the classical RF or the surround, and regardless whether the preferred stimulus is preferred due to its location in the RF or its orientation. Importantly, they found that selectivity and suppression strength are not correlated, and an interaction between selectivity and suppression strength is needed to predict the attention modulation strength, which could be accounted for by a fitting a model where attention has a multiplicative effect on both the excitatory input and the normalization strength of the attended stimulus. This showed that the effect of attention is the same for classical receptive fields as the surround receptive field.

Major issues:

The authors claim that the attention effect solely depends on the differential response of the component stimuli, regardless of whether the difference arises from spatial selectivity or feature selectivity. They show the former, i.e. that their findings of interaction between selectivity and suppression etc. can be reproduced by looking only at conditions where the same stimulus was presented, so that only spatial selectivity can evoke the observed attention effects. But can you show from your data that feature selectivity alone can reproduce your results (e.g. look at conditions where neuron has low spatial selectivity, i.e. the neurons response is the same if you present the same stimulus at one of two locations; or by pairwise comparisons of different stimuli at the same locations)?

Reviewer #2:

Verhoef and Maunsell recorded the activity of V4 neurons in alert and behaving monkeys performing a dot-detection task requiring focused spatial attention. The authors were interested in comparing attentional modulation under conditions where two Gabor stimuli were placed within the classical receptive field versus conditions where one Gabor was placed within the classical receptive field and the second Gabor was placed in the adjacent region of the extraclassical surround. They observed a variety of attentional modulations in both conditions and they found that attentional modulation correlates with stimulus selectivity and also with stimulus-induced suppression in both conditions. Importantly, the relationships between attentional modulation and stimulus selectivity and induced suppression were complicated as stimulus selectivity and induced suppression were not themselves correlated among the recorded neuronal population. Instead, attentional modulation was greatest when selectivity was high and stimulus-induced suppression was also high. The authors then proposed a normalization model that nicely accounts for these relationships. Two aspects of this study were particularly novel and interesting.

First, in describing how variability in attentional modulation of neuronal activity relates to neuronal stimulus selectivity and stimulus-induced suppression, the authors add to general knowledge about attentional modulation of neuronal activity. The use of the normalization model to explain these relationships is elegant.

Second and related, the authors examine the relative contributions of stimulus selectivity and stimulus-induced suppression toward overall attentional modulation, which is also supported by the normalization model.

While these findings are important and advance the field of attention, there are some concerns about the context in which these results are presented and also some technical concerns that slightly dampen overall enthusiasm for the study.

One concern involves placing the results of this study in the context of prior, quite similar work. In the Introduction, the authors state that "the way that attention acts in the cRF versus the surround has not been compared directly" and they cite the work of Sundberg et al. (2009) and Sanayei et al. (2015), both of which examined attentional modulation of responses to visual stimuli within the classical receptive field and the extraclassical surround. In the Sundberg et al. study, the stimuli were similar and similarly placed in the receptive field relative to the current study. The authors' statement therefore seems inaccurate. But additionally, the authors never directly compare their findings with those of Sundberg et al. and Sanayei et al. If the authors are going to make the case that their work overturns or expands upon that of prior groups, then this needs to be explicitly described in the Introduction and/or Discussion.

A second concern involves the amount of suppression observed in the two Gabor stimulus configurations. In the Abstract and Introduction the authors relate their work to the rich literature on relative contributions of classical and extraclassical regions of the receptive field. The data in Figure 4 indicate that there is actually very little surround suppression observed in the sRF-cRF condition. The authors are aware of this and use careful wording (stimulus-induced suppression) and they include Figure 7 to illustrate that the stimulus-induced suppression they observe is qualitatively similar to surround suppression measured previously. But throughout the study, the results are consistent across the cRF-cRF and sRF-cRF conditions. The authors interpret this finding as evidence that attention acts uniformly across both the center and surround of the receptive field. But another possible explanation is that surround suppression was never in fact activated by the stimulus configurations used and thus contributions of the extraclassical surround were not accurately assessed.

There are two possible explanations for the lack of surround suppression in their data that the authors should address. First, the stimulus configuration of two adjacent Gabors may not be ideal for activating surround suppression in V4 neurons because the stimuli are not covering large enough extents of the center or surround. Second, and perhaps more importantly, if recordings are multi-unit rather than single-unit measurements, this will underestimate the extent of surround suppression. The electrodes used in this study are similar to those used previously by the Maunsell group and prior studies often reported data from multi-unit rather than single-unit recordings. Low impedance electrodes, such as those reported here, are not optimal for recording well-isolated single units. Quantities of well-isolated single-unit versus multi-unit recordings used for the analyses are not reported. Given that surround suppression will vary depending on the type of neuronal activity recorded, it would be helpful to see an example of well-isolated single units and to have a quantification of the number of single-unit versus multi-unit recordings that were utilized for each measurement involving stimulus-induced suppression.

Reviewer #3:

Verhoef and Maunsell measure responses of V4 neurons under conditions in which number of stimuli (2D Gabor patches), stimulus orientation, stimulus position, and spatial attention vary. Their main empirical finding is that attention modulation (change in response when attention is shifted between two stimuli in or near the receptive field) is a function not only of selectivity (difference in response to the two stimuli presented individually) but also suppression (reduction of response to the preferred stimulus when the non-preferred stimulus is added) (Figure 4). Moreover, they argue that selectivity and suppression interact to determine the strength of attention effects, and that attention modulation is weak or absent if either is low (Figure 5). They fit a model incorporating selectivity and suppressive effects to each neuron. Based on the fitting results, they argue that the strength of attentional modulation is entirely determined by selectivity combined with suppression, and that attention is a uniform multiplier interacting with these terms.

The dependency of attentional modulation on suppression looks clear, especially for interactions between one stimulus in the classical receptive field (CRF) and another stimulus in the surround. I believe this is a novel and important finding. The supra-additive interaction between selectivity and suppression is visually clear from Figure 5, but I had trouble understanding the analytical support for this. I think the authors performed analyses specific to cases with 0 selectivity and cases with 0 suppression, but this is not clear from the manuscript. In one case they report highly significant p values, in the other non-significant, so it is not clear how the analysis demonstrates supra-additivity.

The main claim of the paper is that attention acts in a uniform way on all stimuli and all CRF and surround positions, in that attentional modulation is entirely predictable from stimulus selectivity and location-specific suppression values. This claim is based on the cross-validation performance of models fit to each neuron with a single term for attentional modulation but multiple terms for stimulus responses and suppression effects. I think there are a number of problems with this analysis that undercut the main claim of the paper.

First, the design of the model seems arbitrary and inconsistent with the findings and the Discussion. One of the suppression values in the denominator, alpha1, is always set to 1. I assume that L1 and alpha1 pertain to the preferred or driving stimulus in a given stimulus pair, at least in most cases, although this is not clear. The Discussion mentions that attentional modulation depends in part on the suppression value at the location of the driving stimulus. How could the model capture that value if it is always fixed? Alpha2 can vary, and must be near 0 for cases where suppression is low (in order to produce equivalent responses when the non-preferred stimulus is added). But if alpha1 is always equal to 1, a low alpha2 value actually contributes to larger attentional modulation, i.e. a larger difference between response with attention at location 1 and response with attention at location 2. This would be the opposite of the empirical finding that low suppression results in low attentional modulation. Some of these issues might be clarified with a more complete presentation of the model and how it is applied to different cases.

Second, it is not clear from this manuscript that the solutions found by the model are reasonable. There are no examples or population plots to show how fitted parameters explain responses under different conditions and how they vary across neurons to explain different experimental results. The only information about fitted parameter values is in Figure 7, which shows that fitted α values are tightly clustered around 0.5. This seems incompatible with Figure 4, showing that measured suppression values cluster near 0. The authors need to show how their model works to explain the different attention effects they measured.

Third, the results of the modeling analyses are presented only in terms of average variance explained in the response pattern, without any specific analysis of how well attentional modulations are predicted. The variance explained values are extremely high, but I would guess this is because most of the response variance depends on differences in stimulus type and location (rather than attention), and the model fits a separate excitatory value for each stimulus type/location combination and a separate suppression value for each location. With all these separate terms, high variance explained seems guaranteed, even with cross-validation, since fitting all these terms requires that the training set contains all stimulus type/location combinations and all suppression locations. (The manuscript does not specify how training and testing conditions were divided.) The variance due to attention is a small fraction of overall response variance, less than 10% based on Figure 6 (and on average modulation at median suppression shown in 4E,F), and variability in attention effects across stimulus conditions looks even smaller. Thus, in most cases the individual models could get attention effects completely backwards and still produce high average variance explained values. That seems quite possible, since the fitting procedure will be driven primarily by the large variance between stimulus conditions. The authors effectively confirm this by showing that the fitted attention parameter is unnecessary for the high variance explained results.

In summary, the demonstration that attention modulation depends on suppression seems clear and noteworthy. However, the main claim of the paper, that attentional modulation is uniform, depending only on selectivity and suppression, would need a different analysis directed specifically at attentional effects as opposed to overall response variance across many different stimulus conditions.

[Editors’ note: what now follows is the decision letter after the authors submitted for further consideration.]

Thank you for resubmitting your work entitled "Attention operates uniformly throughout the Classical Receptive Field and the Surround" for further consideration at *eLife*. Your revised article has been favorably evaluated by David Van Essen (Senior editor), a Reviewing editor, and three reviewers.

The manuscript has been improved, but a number of remaining issues that need to be addressed before acceptance, as detailed below:

Reviewer #1:

"Attention operates uniformly throughout the classical receptive field and the surround."

In general, I think the authors have addressed the reviewer's comments clearly and compellingly. However, a few issues remain unclear:

1) The authors explain their results in relation to Sundberg as arising from the fact that the latter did not vary the position of the second stimulus: "Consequently, both the selectivity and the stimulus-induced suppression index varied with a single variable, the strength of the second stimulus relative to the strength of the fixed stimulus." This is not clear. Even at a fixed location, it seems L1, L2, alpha1, and alpha2 are independent in theory, and selectivity and suppression could be uncorrelated. Furthermore, as shown in Figure 7, excitation and suppression fall off at independent rates with distance from RF center, and presumably Reynolds sampled a variety of locations wrt RF center. Thus the authors need to clarify and explain precisely the statement in the quotation marks above (preferably with reference to equations). Are the authors claiming that Eq1 logically implies that suppression and selectivity should be correlated when measured in the manner of Reynolds? Related to this, it would be helpful if the authors actually analyzed the subset of their data corresponding to Reynold's paper, and show that they can reproduce the correlation between suppression and selectivity of that paper.

2) I did not see any response to Reviewer 3's point: "But if alpha1 is always equal to 1, a low alpha2 value actually contributes to larger attentional modulation, i.e. a larger difference between response with attention at location 1 and response with attention at location 2. This would be the opposite of the empirical finding that low suppression results in low attentional modulation." Please address.

Reviewer #3:

Original review: "The dependency of attentional modulation on suppression looks clear […] demonstrates supra-additivity."

The authors added an explanatory clause to the main text, but the supporting analyses remain unclear. The reported p-values are for "main effects at 0 selectivity/suppression". What does "at 0" mean? Is it an analysis performed on a subset of cells with no significant selectivity/suppression? Or is it an analysis performed on a row or column (how wide?) from the average plot? If so, how many neurons actually occupied those rows or columns?

The other confusing thing here is that p values for significance of main effects are given as though they support the statements about the weak effects, which they don't, in the sense that the p-values are actually most significant for main effects of selectivity, which are being discounted as weak. The discrepancy needs explanation. There must be a more appropriate numerical basis for the statement that both selectivity and suppression are required for strong attention effects.

In addition, it is hard to reconcile Figure 5, where attention effects are strongly focused near selectivity = 1 (even if you collapse across suppression) with Figure 4, where attention effects are equally strong at selectivity = 0.5. What explains the difference?

Original review: "The main claim of the paper is that attention acts in a uniform way on all stimuli […] claim of the paper."

The authors respond that, in addition to the model, the claim is based on the visual similarity of Figure 5, and on "the statistical tests that we performed on these neuronal responses (see the discussion on the general linear model above)". 5A and B do not look entirely similar; in 5A the peak is elongated horizontally, showing that attention is more sensitive to suppression when the distractor is in the receptive field; in 5B the peak is elongated vertically, showing that attention is more sensitive to selectivity when the distractor is in the surround. A direct test of whether the attention effects are equivalent in 5A and 5B would be some kind of two-dimensional Kolmogorov-Smirnov analysis (see, e.g., Lopes, Reid, Hobson, Proceedings of Science, XI International Workshop on Advanced Computing and Analysis Techniques in Physics Research April 23-27 2007 Amsterdam, the Netherlands.) The linear regression is a less direct way of testing the question, and I don't think that it could capture the differences between peak shapes in 5A and 5B, in which case the new Bayes Factor analysis would not be meaningful.

Original review: "First, the design of the model seems arbitrary and inconsistent with the findings and the Discussion. […] Some of these issues might be clarified with a more complete presentation of the model and how it is applied to different cases."

The authors explain that it is only the relative values between parameters that matter, and support this by writing out a multiplication of all terms by 1/alpha1. This still doesn't make sense to me in that the α values need to be able to go to 0, as in Figure 5/Figure 5—figure supplement 1, to represent absence of suppression. If suppression due to stimulus 1 is entirely absent (in which case the multiplier 1/alpha1 would be undefined), how would the model capture that? The addition of Figure 5/Figure 5—figure supplement 1 is certainly helpful in understanding the operation of the model, but it doesn't solve this confusion for me.

Original review: "Second, it is not clear from this manuscript that the solutions found by the model are reasonable. […] The authors need to show how their model works to explain the different attention effects they measured."

In their response, the authors provide an extended explanation of how α values can be balanced out in either direction by L values, concluding that "in general, there exists no direct relationship between the suppression-index in Figure 4 and the α term." To the extent this is true, Figure 7 and the accompanying legend are highly confusing, since they equate α with suppression in the labels and in the text. The fact that the new supplementary example with 0 suppression also has a 0 α value will tend to add to this confusion. Since they are plotting α in Figure 7, the authors need to explain what it means and how it relates (if at all) to suppression, not just to reviewers but also to readers.

Original review: "Third, the results of the modeling […] The authors effectively confirm this by showing that the fitted attention parameter is unnecessary for the high variance explained results."

The authors correctly point out that the models must not be failing to capture attention effects given the close approximation to population level effects in Figure 4. And, they reiterate that removing the attention term reduces overall explained variance from 87% to 79%. My main point had simply been that these were both extremely high values, and this is due to the fact that the number of terms in the model is on the same order as the number of conditions, so a close fit is not surprising or informative in either case. The cross-validation does not change this because, given the low number of conditions and the description now given in methods, it amounts to simply splitting repetitions from identical trial types into two groups and proving they give the same result. I don't think the 87% explained variance with the full model or the 8% drop in explained variance by themselves establish that "attention acts uniformly across the cRF and the surround". (The exceedingly close fits are due to the similar number of terms and conditions; the 8% drop shows that the attention term mattered, but it doesn't address how exactly attention behaved.) Nor, as explained above, do I think this is established by visual similarity between Figure 5 or the linear regression analysis. As stated above, I think the direct test of this proposition would be a statistical comparison of 5A and 5B. Regardless of the result, the clear and more interesting result here is the criticality of the interaction between selectivity and suppression for attention.

[Editors' note: further revisions were requested prior to acceptance, as described below.]

Thank you for resubmitting your work entitled "Attention operates uniformly throughout the Classical Receptive Field and the Surround" for further consideration at *eLife*. Your revised article has been favorably evaluated by David Van Essen (Senior editor) and the Reviewing editor.

The manuscript has been improved and we greatly appreciate the detailed, equation-based clarifications in response to several of the points raised by the reviewers.

However, the reviewers remain concerned whether the authors have really provided evidence that attention acts uniformly throughout the receptive field. The reviewers would like clarification of the following points:

1) Is there any reason that 5A and 5B should look identical? In the initial reviewer response, the authors stated, "We would like to emphasize that the claim that attention acts in a uniform way on all cRF and surround positions is supported by more than just the model that accurately accounts for all neuronal responses, irrespective of the receptive field position and attention condition. In particular, this uniformity can be seen directly in the similarity of the neuronal responses in Figure 5, which show that the dependency of attention modulation on selectivity and suppression does not depend on whether stimuli are presented in the cRF or the surround." The reviewers see a clear difference between 5A and B. This seems to provide evidence that attention is more critically dependent on selectivity in the CRF-CRF condition, and more critically dependent on suppression in the CRF-SRF condition; quite the opposite of operating uniformly. The reviewers are confused whether the authors have a hypothesis or explanation for why the dependency of attention on suppression and selectivity should differ between CRF (5A) and SRF (5B); it seems that the difference is contrary to the authors' conclusion. If they do have a hypothesis about the difference, please state it and test it, and modify the "uniform operation" conclusion accordingly.

2) In the most recent reviewer response, the authors seem to be backing away from claim that 5A and B need to look similar, and are appealing to the similarity between the model fits (5D,E) and the data (5A, B). The reviewers remain concerned that the reason why the model fits assuming uniform attention are so good is that all of the non-attention related terms (separate excitatory term for each stimulus type/location combination and a separate suppression value for each location) are doing all the work. Since the number of terms being fitted is nearly equal to the number of measurements, this guarantees over fitting, and it can't be cured by cross-validation because every condition would need to be in both portions of the data to fit all the stimulus- and position-specific terms.

It would seem advisable to include an explicit caveat about this.

3) Most importantly, the obvious way to test their claim is some kind of direct comparison of attentional modulation between CRF and SRF. Can the authors do this?

4) Have the authors tried to fit other models with fewer non-attention related variables (e.g., keeping suppression constant)? How does this affect the attention term?

Overall, the reviewers would be most convinced by a new analysis directly showing that attentional modulation as a function of selectivity and suppression is the same in the cRF and surround, without appeal to goodness of model fits. Without such an analysis, it remains unclear whether the authors have identified a truth about the brain, that attention acts uniformly, or whether they have simply shown that a viable model of neurons can be constructed in which attention acts uniformly, but other models with a variable attention term are equally plausible.

---

## [Author Response]

[Editors’ note: the author responses to the first round of peer review follow.]

*Major issues:*

*The authors claim that the attention effect solely depends on the differential response of the component stimuli, regardless of whether the difference arises from spatial selectivity or feature selectivity. They show the former, i.e. that their findings of interaction between selectivity and suppression etc. can be reproduced by looking only at conditions where the same stimulus was presented, so that only spatial selectivity can evoke the observed attention effects. But can you show from your data that feature selectivity alone can reproduce your results (e.g. look at conditions where neuron has low spatial selectivity, i.e. the neurons response is the same if you present the same stimulus at one of two locations; or by pairwise comparisons of different stimuli at the same locations)?*

Thanks for the suggestions. We repeated the same analysis on the data from conditions with low spatial selectivity. This analysis confirmed that feature (orientation) selectivity alone can reproduce our results. When attention shifts between two approximately equally responsive cRF positions (less than 2 spike/s response difference when each of two cRF positions is stimulated with an identical single stimulus), similar effects were observed (main effect of feature selectivity: p<0.001; main effect suppression: p=0.7; interaction between feature selectivity and suppression: p=0.02; multiple linear regression).

This result is further corroborated by the modeling results, as the model makes no distinction between spatial and feature selectivity, yet it fits the data well.

This new result has been added to the findings on (Results section, subsection “A spatially-tuned normalization model captures attention modulation inside the cRF and in the surround”): "Similar to these previous studies, we reproduced the above findings using the data from conditions with low spatial selectivity. When attention shifts between stimuli at two approximately equally responsive cRF positions (less than 2 spike/s response difference when each of two cRF positions is stimulated with an identical single stimulus), similar effects were observed (main effect of feature selectivity: p<0.001; main effect suppression: p=0.7; interaction between feature selectivity and suppression: p=0.02; multiple linear regression). Next, we examined whether the converse situation, i.e. same stimuli at unequally-responsive cRF positions, would produce attention modulations comparable to those described earlier..…"

*Reviewer #2:*

*Verhoef and Maunsell recorded the activity of V4 neurons in alert and behaving monkeys performing a dot-detection task requiring focused spatial attention.*

[…]

*But additionally, the authors never directly compare their findings with those of Sundberg et al. and Sanayei et al. If the authors are going to make the case that their work overturns or expands upon that of prior groups, then this needs to be explicitly described in the Introduction and/or Discussion.*

Both Sundberg et al. (2009) and Sanayei et al. (2015) used conditions with one stimulus inside the classical receptive field and at least one other stimulus in the surround (sRF-cRF condition in our study). However, neither of these studies compared responses to a condition in which both stimuli were positioned inside the cRF (cRF-cRF condition).

Hence, these investigators could not examine how attention affects neuronal responses when shifted between two stimuli inside the cRF (cRF-cRF condition) versus when shifted between one stimulus inside the cRF and another stimulus inside the surround (sRF-cRF condition). This comparison is crucial for determining whether attentional effects on neuronal responses differ between the cRF and the surround. Additionally, we show the similarity between attentional effects in the cRF and the surround can only be revealed by examining the interaction between stimulus selectivity and stimulus-induced suppression (Figure 4). No previous study has ever done this.

Finally, Sundberg et al. did not test whether a single model could fit data from both the cRF and the surround. In our study, the good fits of the spatially-tuned normalization model to the data obtained in both receptive field configurations provided critical corroborating evidence for the conjecture that attention acts similarly inside the cRF and the surround. Sanayei et al. fit different (normalization) models, but never measured the neuronal effects of attention when attention was directed to the surround stimulus; they only compared the effects of attention to the cRF stimulus, with or without surround stimulation, versus attention to a distant stimulus. Attention was never directed to the surround stimuli. Thus, Sanayei et al. lacked the crucial information needed to examine how surround suppression affects attention modulation and to test the efficacy of normalization models.

We believe that most of the confusion comes from our wording, which did not mention that attention should shift between stimuli either presented inside the cRF or between a cRF and surround stimulus. We now write on (third paragraph of the Introduction section): "For example, the way that attention acts on neuronal responses when shifted between stimuli inside the cRF versus when shifted to stimuli inside the surround has not been compared directly16,23,29."

We now also discuss both studies in the Discussion on (fourth paragraph of Discussion section): "Both Sundberg et al. (2009) and Sanayei et al. (2015) used conditions with one stimulus inside the classical receptive field and at least one other stimulus in the surround (sRF-cRF). However, neither of these studies used a condition with both stimuli positioned inside the cRF (cRF-cRF). Hence, a direct comparison of attention modulation within the cRF and the surround could not be performed in these studies. This comparison is crucial to determine whether the neuronal effects of attention differ between the cRF and the surround. Importantly, we found that seeing the similarity between attentional effects in both receptive field regions requires examination of the combined relationship, i.e. interaction, between attention modulation and both stimulus selectivity and stimulusinduced suppression (Figure 4 versus 5). The interaction between these variables was similar in both receptive field configurations.

Sanayei et al. fit different (normalization) models, but never measured the neuronal effects of attention when attention was directed to the surround stimulus; the authors only compared the effects of attention to the cRF stimulus, with or without surround stimulation, versus attention to a distant stimulus. Attention was never directed to the surround stimuli. Thus, Sanayei et al. lacked the crucial information needed to examine how surround suppression affects attention modulation and to test the efficacy of normalization models. We tested whether a single model could fit both the cRF-cRF and sRF-cRF data. The good fits of the spatially-tuned normalization model to the data obtained in both receptive field configurations provided further evidence that attention acts similarly inside the cRF and the surround."

*A second concern involves the amount of suppression observed in the two Gabor stimulus configurations.*

[...]

There are two possible explanations for the lack of surround suppression in their data that the authors should address. First, the stimulus configuration of two adjacent Gabors may not be ideal for activating surround suppression in V4 neurons because the stimuli are not covering large enough extents of the center or surround.

Our visual stimuli were not designed for maximal surround suppression, which would have required filling the surround with a high contrast stimulus. This was not possible in our experiment, which required attention that was comparably restricted in space in all conditions (cRF-cRF and sRF-cRF). It is likely that surround suppression was limited primarily because of the limited extent of the Gabor stimuli. However, we know of no reason to believe that the Gabors led to qualitative (rather than quantitative) differences from what would have been obtained had optimal surround activation been possible. We observed very similar surround suppression as that observed in previous studies on attention in the surround (Sundberg et al., 2009; Sanayei et al., 2015).

Although the mean surround suppression was modest, a substantial proportion of the neurons was significantly suppressed by the surround stimulus (see Figure 4 black bars, p<0.01).

Using Gabors made it possible for us to explore activation of the surround at varying offsets from the RF center. Figure 7 shows that surround suppression decreases with increasing distance from the RF center. Including data from near and far portions of the surround allowed us to show that attention acts comparably across the range of distances. As noted in a response to Reviewer 1 (above), limiting our analyses to sites with statistically significant suppression indices did not change the results. The consistency across the cRF-cRF and sRF-cRF conditions also holds when considering only neurons with strong surround suppression (Figure 5). We now also present some more single-neuron examples that show clear surround suppression (see below and also Figure 4—figure supplement 1 and Figure 3).

Hence, although larger surround stimuli would have caused more suppression, the surround stimuli in our study produced clear suppression and attention interacted with this surround suppression as it did with suppression from the cRF.

*Second, and perhaps more importantly, if recordings are multi-unit rather than single-unit measurements, this will underestimate the extent of surround suppression. The electrodes used in this study are similar to those used previously by the Maunsell group and prior studies often reported data from multi-unit rather than single-unit recordings. Low impedance electrodes, such as those reported here, are not optimal for recording well-isolated single units. Quantities of well-isolated single-unit versus multi-unit recordings used for the analyses are not reported. Given that surround suppression will vary depending on the type of neuronal activity recorded, it would be helpful to see an example of well-isolated single units and to have a quantification of the number of single-unit versus multi-unit recordings that were utilized for each measurement involving stimulus-induced suppression.*

All results reported in this study were based on single units, not multi-unit sites.

However we did observe very similar results when analyzing the multi-units. We apologize for this point not being clear in the previous version of the manuscript. We have added on (Third paragraph of Introduction section): "All results presented here are based on the activity of these 728 single neurons, but all findings were confirmed in the responses of 12067 multi-unit clusters (M1: 4709; M2: 7358)." We also added the waveforms of the recorded neurons (blue) plus that of the multi-unit activity measured at the same electrode (grey) in Figure 3.

Finally, we added a new supplemental figure in which we present some wellisolated single units that demonstrate strong and significant surround suppression (see Figure 3 for another example). We have added on Results section, subsection “Relationship between selectivity, stimulus-induced suppression and attention modulation”: "The black bars in Figure 4 represent neurons that were significantly (p<0.01) suppressed by the non-preferred (surround) stimulus. See Figure 4—figure supplement 1 for some example neurons with significant surround suppression (see also Figure 3)."

*Reviewer #3:*

*Verhoef and Maunsell measure responses of V4 neurons under conditions in which number of stimuli (2D Gabor patches), stimulus orientation, stimulus position, and spatial attention vary. Their main empirical finding is that attention modulation (change in response when attention is shifted between two stimuli in or near the receptive field) is a function not only of selectivity (difference in response to the two stimuli presented individually) but also suppression (reduction of response to the preferred stimulus when the non-preferred stimulus is added) (Figure 4). Moreover, they argue that selectivity and suppression interact to determine the strength of attention effects, and that attention modulation is weak or absent if either is low (Figure 5). They fit a model incorporating selectivity and suppressive effects to each neuron. Based on the fitting results, they argue that the strength of attentional modulation is entirely determined by selectivity combined with suppression, and that attention is a uniform multiplier interacting with these terms.*

*The dependency of attentional modulation on suppression looks clear, especially for interactions between one stimulus in the classical receptive field (CRF) and another stimulus in the surround. I believe this is a novel and important finding. The supra-additive interaction between selectivity and suppression is visually clear from Figure 5, but I had trouble understanding the analytical support for this. I think the authors performed analyses specific to cases with 0 selectivity and cases with 0 suppression, but this is not clear from the manuscript. In one case they report highly significant p values, in the other non-significant, so it is not clear how the analysis demonstrates supra-additivity.*

We used a general linear regression model to examine whether selectivity and suppression interact to determine the magnitude of attention modulation. This regression model is explained in the Methods. The model includes two main effects, one for selectivity and one for suppression, in addition to an interaction term.

attention modulation = selectivity ⋅β1+ suppression ⋅β 2+ selectivity ⋅suppression ⋅β3+error

In this model, the main effect of selectivity measures the contribution of selectivity to attention modulation given that suppression is zero. Similarly, the main effect of suppression measures the contribution of suppression to attention modulation given that selectivity is zero. For example, if suppression is zero, the suppression and interaction term (selectivityÅ~suppressionÅ~β3) both go to zero, leaving only the selectivity term β1, which specifies the contribution of selectivity to attention modulation. Conversely, if selectivity is zero, the only non-zero term is the β2 term, which specifies the contribution of suppression to attention modulation.

The main effect of suppression was never significant. The main effect of selectivity was, but the effect was very small as can be seen in Figure 5. These main effects are not important for the message in this manuscript, but are provided for completeness. In striking contrast, all terms corresponding to the interaction between selectivity and suppression (β3) were highly significant in both the cRF-cRF and the sRF-cRF configuration, and for each monkey separately. This demonstrates non-additivity (Results section, subsection “Stimulus selectivity and stimulus-induced suppression interact in determining attention modulation and do so similarly inside the cRF and the surround”).

We show that this non-additivity is similar in each receptive field configuration (using a 3-way interaction between selectivity, suppression and receptive field configuration,), meaning that β3 (but also β1 and β2) does not differ significantly between the cRF-cRF and sRF-cRF condition. Because a non-significant effect does not indicate the absence of an effect, we performed a new Bayesian general linear model analysis. This analysis showed that the observed data are 347 times more likely to agree with a regression model that does not distinguish between the cRF-cRF and sRF-cRF configurations than with a model that does include RF-configuration as a predictor. This means that attention modulation is driven by similar mechanisms within the cRF and the surround.

We now clarify this (same section)"Importantly, in each RF configuration the regression model also included an interactive product term, which measured the dependency of attention-related modulation on both selectivity and stimulus-induced suppression, i.e. this term measures whether the relationship between selectivity, suppression and attention modulation is non-additive (see Methods for further information)."

We also expanded the regression description in Analyses subsection of the Methods section:

"We used multiple linear regression to examine if attention-related modulation depends on the interaction between stimulus selectivity and stimulus-induced suppression.

[…]

Conversely, if selectivity is zero, the only non-zero term is the β2 term, which specifies the contribution of suppression to attention modulation."

*The main claim of the paper is that attention acts in a uniform way on all stimuli and all CRF and surround positions, in that attentional modulation is entirely predictable from stimulus selectivity and location-specific suppression values. This claim is based on the cross-validation performance of models fit to each neuron with a single term for attentional modulation but multiple terms for stimulus responses and suppression effects. I think there are a number of problems with this analysis that undercut the main claim of the paper.*

We would like to emphasize that the claim that attention acts in a uniform way on all cRF and surround positions is supported by more than just the model that accurately accounts for all neuronal responses, irrespective of the receptive field position and attention condition. In particular, this uniformity can be seen directly in the similarity of the neuronal responses in Figure 5, which show that the dependency of attention modulation on selectivity and suppression does not depend on whether stimuli are presented in the cRF or the surround. In addition, the claim of uniformity is supported by the statistical tests that we performed on these neuronal responses (see the Discussion on the general linear model above). The good model fits provide additional support for this claim.

*First, the design of the model seems arbitrary and inconsistent with the findings and the Discussion. One of the suppression values in the denominator, alpha1, is always set to 1. I assume that L1 and alpha1 pertain to the preferred or driving stimulus in a given stimulus pair, at least in most cases, although this is not clear. The Discussion mentions that attentional modulation depends in part on the suppression value at the location of the driving stimulus. How could the model capture that value if it is always fixed? Alpha2 can vary, and must be near 0 for cases where suppression is low (in order to produce equivalent responses when the non-preferred stimulus is added). But if alpha1 is always equal to 1, a low alpha2 value actually contributes to larger attentional modulation, i.e. a larger difference between response with attention at location 1 and response with attention at location 2. This would be the opposite of the empirical finding that low suppression results in low attentional modulation. Some of these issues might be clarified with a more complete presentation of the model and how it is applied to different cases.*

We are sorry about the confusion and address these concerns here.

First, we should point out that in the cRF-cRF condition both stimuli are typically driving stimuli, meaning that both stimuli contribute a non-zero L-term.

Second, we used multi-electrode array recordings, recording simultaneously from many neurons. For some neurons the stimulus at position 1 (and L1 and α1) was the stronger driver (e.g., positioned closer the receptive field center), while for other neurons the stimulus at position 2 drove the neuron stronger (larger L2 term). For the sRF-cRF condition, position 1 was the surround position for some neurons, while position 2 was the surround position for other neurons. This depended on where the two stimulus positions fell relative to the receptive field of each neuron, which varied across days because we recorded different neurons and used different stimulus positions on different days (Materials and methods section, subsection “Analyses”).

Third, α1 was set to 1 only to simplify the model. Letting it vary has no effect. If it is allowed to vary, the resulting fit can always be converted to a fit with α1 =1 by multiplying all terms by the appropriate factor, this can be seen as follows:R1,2=L1+L2α1+α2+σ=(1α11α1)L1+L2α1+α2+σ=L1α1+L2α11+α2α1+σα1=L1'+L2'1+α2'+σ'

Transforming α1 to 1 causes the other parameters to take different values (indicated by primes in the equation) but has no other consequence. In general, the absolute values of the model parameters have no meaning. Only the relative values of the parameters with respect to each other are important.

For the spatially-tuned normalization model, fixing α1 to 1 provides the benefit of placing all the other model parameters on a similar scale, where they can be more readily compared. For example, if receptive field position 2 has an α2 > 1, it means that stimuli presented at position 2 will contribute more suppression than stimuli presented at position 1. If α1 were free to vary, α2 could not be interpreted based on its absolute value alone. The crux of the model lies in the relative values of its parameters, which vary spatially (Figure 7). Below we also present some single-neuron examples to further explain the model.

*Second, it is not clear from this manuscript that the solutions found by the model are reasonable. There are no examples or population plots to show how fitted parameters explain responses under different conditions and how they vary across neurons to explain different experimental results. The only information about fitted parameter values is in Figure 7, which shows that fitted α values are tightly clustered around 0.5. This seems incompatible with Figure 4, showing that measured suppression values cluster near 0. The authors need to show how their model works to explain the different attention effects they measured.*

Figure 7 shows how the key model parameters (the α- (grey) and L- (black) parameters) vary across the receptive field. As described in the manuscript, variation in the other model terms is less important. For example, (Results section, subsection “Spatial variability in excitation and suppression underlies differencs in attention modulation across neurons”) we point out that variability in the attention parameter β is inconsequential because a model with a fixed β-parameter fits the data equally well. Note that this does not mean that the betaparameter is unimportant. In fact, the β-parameter is crucial to capture the neuronal effect of attention. It does mean that the β-parameter does not need to vary across different neurons to account for the data (see below). Finally, the values of the σ parameter tend to cluster around zero. We now give information on the σ-values on (see also the response to reviewer 2).

It is important to realize that the suppression values in Figure 4 are not directly related to the α values in Figure 7. The suppression values in Figure 4 are a consequence of a balance of excitation and suppression. In the model, excitation is represented by the L-parameter and suppression is represented by the α-parameter.

For example, if one presents a new stimulus adjacent to an already presented stimulus, the neuronal response might decrease, and a positive stimulus-induced suppression index will be measured. The response decrease means that the added stimulus induced more suppression than excitation, but this can happen with different values of the α- and Lparameter.

For example the L-term might be large and the α-term even larger, or the

L-term might be minimal with an α-term that is small but large enough to suppress the response.

An α value of 0.5 will suppress neuronal responses when the corresponding Lterm is small, but the same α value might lead to increased responses when the corresponding L-term is large. This is why stimuli in the surround often suppress neuronal responses: surround stimuli contribute little excitation relative to suppression (low L-term compared to α-term). However, those same stimuli might readily increase the response when presented inside the receptive field center (high L-term compared to α-term).

The near-zero values in Figure 4 show that a second stimulus often produces small changes in the balance between excitation and suppression. Nevertheless, as can be seen in Figure 4, in many cases stimuli did tip the balance between excitation and suppression in favor of suppression (positive suppression indices) or excitation (negative suppression values).

Thus the stimulus-induced suppression indices depend on the balance between excitation and suppression, which is modeled by the relative values of the L- and alphaterm.

In general, there exists no direct relationship between the suppression-index in

Figure 4 and the α-term. Below we also present some single-neuron examples to further explain the model.

*Third, the results of the modeling analyses are presented only in terms of average variance explained in the response pattern, without any specific analysis of how well attentional modulations are predicted. The variance explained values are extremely high, but I would guess this is because most of the response variance depends on differences in stimulus type and location (rather than attention), and the model fits a separate excitatory value for each stimulus type/location combination and a separate suppression value for each location. With all these separate terms, high variance explained seems guaranteed, even with cross-validation, since fitting all these terms requires that the training set contains all stimulus type/location combinations and all suppression locations. (The manuscript does not specify how training and testing conditions were divided.) The variance due to attention is a small fraction of overall response variance, less than 10% based on Figure 6 (and on average modulation at median suppression shown in 4E,F), and variability in attention effects across stimulus conditions looks even smaller. Thus, in most cases the individual models could get attention effects completely backwards and still produce high average variance explained values. That seems quite possible, since the fitting procedure will be driven primarily by the large variance between stimulus conditions. The authors effectively confirm this by showing that the fitted attention parameter is unnecessary for the high variance explained results.*

It is not true that the attention parameter (β) is unnecessary for the high variance explained. Although an attention parameter that varies across neurons is unnecessary, the attention parameter by itself, with a value fixed across all neurons, is crucial to model any attention modulation. What we show is that the attention parameter does not have to vary across neurons, but its fixed value is pivotal in order to predict any attention effects.

Without the attention parameter the model would not explain any curve in Figure 4, and Figure 5 because it would predict zero attention modulation across the board, in clear contradiction to the data. A model with no β parameter does a significantly poorer job (median explained 2-fold cross-validated variance of 79% versus 87% for the model with β parameter; p<0.01).

As the reviewer notes, a substantial amount of variance remains explained because of the variance that arises from stimulus differences. This is also because the responses of some neurons were little affected by attention (see neurons with weak selectivity and suppression in Figure 5), producing small decreases in explained variance when leaving out the β parameter. Other neurons' responses were strongly affected by attention (see neurons with strong selectivity and suppression in Figure 5) and a model with no β parameter explained up to 50% of the variance less than a model with β parameter. Our results show that the magnitude of attention modulation, and thus how much of the response variance is explained by attention and β, varies with suppression and selectivity. Thus the critical test of the model lies in its ability to explain the full range of attention modulations across the range of observed suppression and selectivity. We show that the model does an excellent job of fitting the observed attention modulations across the full range of selectivity and suppression values and in different stimulus and receptive field conditions. Figure 4 plot attention modulation vs. selectivity in both RF configurations. The light grey values in these plots represent the average attention modulations predicted by the model and show that the model precisely captures these trends in the population of neurons across the full range of selectivity values. Similarly Figure 4 show (light grey values) that the model also precisely captures how attention modulation varies with stimulus-induced suppression in the population of neurons across the full range of suppression indices. In addition, Figure 5 show that the model captures attention modulation in the population of neurons within the entire space created by the selectivity and suppression induces, i.e. the model clearly predicts the interaction, and does so very similar to the observed data in Figure 5 . Finally, Figure 6 shows that the model precisely reproduces the average attention modulation within the population of neurons in the conditions where single stimuli are presented at different RF locations. The close correspondence between the model predictions and the observed data are only possible if the model gets the attention effects right.

We now add (Results section, subsection “A spatially-tuned normalization model captures attention modulation inside the cRF and in the surround”): "Figure 4 (light grey points) show that the model precisely accounts for attention modulation across the full range of observed stimulus selectivity and stimulus-induced suppression values, within both the cRF-cRF and sRF-cRF configuration."

We also added a new figure with single-neuron examples to further explain the model and to demonstrate the model's ability to account for attention modulation in the responses of individual neurons (see below, Figure 5—figure supplement 1).

In addition, we now provide details on the cross-validation procedure in the

Methods: "The goodness-of-fit of the model was calculated for each neuron as the percentage explained variance, which was determined by taking the square of the correlation coefficient between the model-predicted responses and the observed neuronal responses across all stimulus conditions. The explained variance was calculated using the average neuronal responses from trials not used to fit the model.

For this purpose we employed two-fold cross-validation, fitting the model based on half of the randomlychosen trials, and using the remaining data to measure the goodness of fit of the model. This procedure was repeated five times, each time using a different random draw, and subsequently averaged across all cross-validations to produce the reported goodness offit."

In sum, the claim that attention acts uniformly across the cRF and the surround is supported by the observed neuronal responses (Figure 5), a statistical analysis of these data (the general linear model, see above), and the spatially-tuned normalization model that accurately accounted for all neuronal responses, irrespective of the receptive field position of the stimuli and the attention condition. We show that this model predicts the observed attention modulation across conditions with one or two stimuli, with stimuli at different RF positions, and across the full range of the observed selectivity and stimulus-induced suppression values. We show this based on the population of neurons (Figure 4, Figure 5, and Figure 6) and based on single-neuron examples (Figure 5—figure supplement 1).

[Editors' note: the author responses to the re-review follow.]

*Thank you for resubmitting your work entitled "Attention operates uniformly throughout the Classical Receptive Field and the Surround" for further consideration at eLife. Your revised article has been favorably evaluated by David Van Essen (Senior editor), a Reviewing editor, and three reviewers.*

*The manuscript has been improved, but a number of remaining issues that need to be addressed before acceptance, as detailed below:*

*Reviewer #1 (:*

*"Attention operates uniformly throughout the classical receptive field and the surround."*

*In general, I think the authors have addressed the reviewer's comments clearly and compellingly. However, a few issues remain unclear:*

*1) The authors explain their results in relation to Sundberg as arising from the fact that the latter did not vary the position of the second stimulus: "Consequently, both the selectivity and the stimulus-induced suppression index varied with a single variable, the strength of the second stimulus relative to the strength of the fixed stimulus." This is not clear. Even at a fixed location, it seems L1, L2, alpha1, and alpha2 are independent in theory, and selectivity and suppression could be uncorrelated. Furthermore, as shown in Figure 7, excitation and suppression fall off at independent rates with distance from RF center, and presumably Reynolds sampled a variety of locations wrt RF center. Thus the authors need to clarify and explain precisely the statement in the quotation marks above (preferably with reference to equations). Are the authors claiming that Eq1 logically implies that suppression and selectivity should be correlated when measured in the manner of Reynolds?*

That is correct: Equation 1 from the spatially-tuned normalization dictates that suppression and selectivity will be correlated when only one of two stimuli is varied. In the spatially-tuned normalization model, the *L* and α terms are indeed independent of each other. However the α terms are fixed at each RF location. Consequently, varying only the stimulus at one RF location, while keeping the other stimulus fixed, corresponds to keeping *L_1_, α_1_* and *α_2_* fixed. Here, *L_1_* is the excitatory drive from the fixed stimulus, *α_1_* is the suppression drive from the fixed stimulus, and *α_2_* is the suppressive drive from stimulating the RF location from the variable stimulus. Thus *L_2_*, the excitatory drive from the variable stimulus, is the only variable that varies when different stimuli are presented at that RF location. The variable *L_2_* term changes both the selectivity and suppression indices in similar ways.

In model terms, this can be seen as:

selectivity=response1−response2response1+response2=L1α1+σ−L2α2+σL1α1+σ+L2α2+σ, where we assume (without loss of generality) that the fixed stimulus at location 1 is the preferred stimulus, i.e. L1α1+σ>L2α2+σ. Also:

Stimulus−induced suppression=response1−response1+2response1+response1+2=L1α1+σ−L1+L2α1+α2+σL1α1+σ+L1+L2α1+α2+σ, where the fixed stimulus at location 1 is again the preferred stimulus.

If we present a weak stimulus at the variable location 2, i.e. *L_2_*≈ 0, the selectivity and stimulus-induced suppression index both become high:

selectivity=response1−0response1+0=L1α1+σ−0α2+σL1α1+σ+0α2+σ=L1α1+σL1α1+σ=1, and

Stimulus−induced suppression=response1−response1+2response1+response1+2==L1α1+σ−L1+0α1+α2+σL1α1+σ+L1+0α1+α2+σ=L1α1+σ−L1α1+α2+σL1α1+σ+L1α1+α2+σThe stimulus-induced suppression index is also high because response1+2=L1+L2α1+α2+σis at its lowest (less subtraction in the numerator) when stimulus 2 adds no excitatory drive to response_1+2_.

Conversely, adding a more potent stimulus at variable location 2, i.e. increasing *L_2_*, decreases the selectivity, because response_2_ is now not zero anymore and thus decreases the numerator. Similarly, the stimulus-induced suppression index will also be smaller, because response1+2=L1+L2α1+α2+σ increases with increasing *L_2_,* which in turn decreases the numerator, L1α1+σ−L1+L2α1+α2+σ, of the stimulus-induced suppression index.

Thus both selectivity and stimulus-induced suppression increase with a weaker stimulus at variable position 2, and both selectivity and stimulus-induced suppression decrease with a more potent stimulus at variable position 2. This shows that, within cells, selectivity and stimulus-induced suppression will be correlated when only one of two stimuli is varied. This agrees with the single neuron examples in Reynolds et al..

Reynolds et al. repeated these measurements for several neurons and included all data points from each neuron into the populations scatter plots. Note that Reynolds et al. always positioned their stimuli at approximately equally-responsive RF locations. The RFs of V4 neurons are approximately Gaussian, which means that equally-responsive RF positions are expected to lie at approximately equal (Mahalanobis) distances from the RF center. Following Figure 7, these equally-responsive RF positions, at equally distant RF positions, are expected to have similar alphas. Because Reynolds et al. only sampled RF locations with similar alphas, similar positive relationships between selectivity and suppression would have been observed for each neuron. Given that all neurons had a similar positive relationship between selectivity and suppression, the population data will also display a similar positive relationship.

This information has been added to the Appendix.

*Related to this, it would be helpful if the authors actually analyzed the subset of their data corresponding to Reynold's paper, and show that they can reproduce the correlation between suppression and selectivity of that paper.*

Our experimental design differed substantially from that of Reynolds et al., because when one stimulus was fixed there were few variants of the other stimulus. For example, when presenting a Gabor with one orientation at RF location 1, our design introduced only two different stimuli at RF location 2: a Gabor with the same orientation or a Gabor with an orthogonal orientation as that at location 1. In contrast, Reynolds et al. used 16 different stimulus conditions.

We compared the two stimulus conditions in which the stimulus at one location was kept constant while the stimulus at the other location varied between two orthogonal orientations. Across neurons and stimuli, we found that the average stimulus-induced suppression index of the stimulus condition with greater selectivity was significantly greater than that of the other stimulus condition with the lower selectivity (stimulus-induced suppression index of 0.07 vs. 0.05, p = 3x10^-13^, paired permutation t-test). Thus despite the limited number of stimulus conditions in our design, we could reproduce the correlation between suppression and selectivity of that paper.

This information has been added to the Appendix.

*2) I did not see any response to Reviewer 3's point: "But if alpha1 is always equal to 1, a low alpha2 value actually contributes to larger attentional modulation, i.e. a larger difference between response with attention at location 1 and response with attention at location 2. This would be the opposite of the empirical finding that low suppression results in low attentional modulation." Please address.*

We apologize for not responding to this point directly (but see Figure 5—figure supplement 1) in the previous revision and will do so here. Our findings show that more suppression from the non-preferred stimulus causes more attention modulation. If *α_1_* (= 1) corresponds to the non-preferred stimulus then a low *α_2_* (i.e. the *α* corresponding to the preferred stimulus) will indeed cause more attention modulation. This is because the non-preferred stimulus has the greatest *α*, so that when the non-preferred stimulus is added to the preferred stimulus it will decrease the neuronal response (*α_1_* is relatively high, but *L_1_* is low), and attention can amplify or weaken this suppression. If the small *α_2_* corresponds to the non-preferred stimulus, there will instead be little attention modulation. Under these circumstances directing attention to stimulus 1 or stimulus 2 does not modulate the response much, i.e. R1att,2 ≈ R1,2att:R1att,2=βL1+L2β+α2+σ≈βL1+0β+0+σ=L11+σ/β≈L11+0=L1R1,2att=L1+βL21+βα2+σ≈L1+β01+β0+σ=L11+σ≈L11+0=L1

Here, we used the fact that the stimulus at position 2 is non-preferred, so *L_2_*≈ 0. Furthermore, using the fact that alpha2 is small as assumed in the question, α2≈ 0. Finally, we used the fact that σis usually small (median σ=0.06; see Methods).

*Reviewer #3:*

*The authors added an explanatory clause to the main text, but the supporting analyses remain unclear. The reported p-values are for "main effects at 0 selectivity/suppression". What does "at 0" mean? Is it an analysis performed on a subset of cells with no significant selectivity/suppression? Or is it an analysis performed on a row or column (how wide?) from the average plot? If so, how many neurons actually occupied those rows or columns?*

We apologize for the confusion. The analyses referred to are based on all data, not a particular selection or subset of the dataset. It is a mathematical property of the linear regression model (in contrast to an ANOVA) with an interaction term that each of the main effects measures the contribution of a variable to the independent variable, given that the other variables (selectivity or suppression) in the model are zero. In our case, the main effect of selectivity gives the rate of change of attention modulation with selectivity, given that suppression is zero. The main effect of suppression gives the rate of change of attention modulation with suppression, given that selectivity is zero. Although the term "main effects" are typically used to denote these effects, "conditional effects" would probably be a better name.

We have added the following to the Methods (Analyses subsection): "Thus the main effects are not estimated from a particular selection or subset of the dataset, but follow mathematically from the linear regression model with interaction term."

*The other confusing thing here is that p values for significance of main effects are given as though they support the statements about the weak effects, which they don't, in the sense that the p-values are actually most significant for main effects of selectivity, which are being discounted as weak. The discrepancy needs explanation. There must be a more appropriate numerical basis for the statement that both selectivity and suppression are required for strong attention effects.*

P-values are of course a different measure than effect sizes and can be very low even for very small effects. As the reviewer notes, the main effect of selectivity in our dataset is a good example of this statement. We included the p-values of the main effects for completeness, although these effects are clearly small (Figure 5). One option would be to mention the regression coefficients in the main text. However, as is known from the statistical regression literature, regression coefficients of main effects become hard to interpret in the context of interactions. This is because the coefficients of the main-effects cannot be interpreted without considering the coefficient of the interaction. Furthermore, the interaction can completely dominate a main effect, because the dependent variable grows non-linearly due to the interaction, as is apparent in our data for high values of the selectivity and suppression indices (upper right corner Figure 5). Thus including the regression coefficients is probably more confusing than elucidating.

We believe that Figure 5 gives the reader the best impression of the effect sizes. We support this impression with the regression analysis, which we believe is an appropriate statistical analysis for these purposes in the sense that it captures a statistically significant non-additive trend in the data.

We now write (Materials and methods section, subsection “Model”: " Specifically, Figure 5 show that when stimulus-induced suppression is low, attention modulation will be weak, even when attention is shifted between a strong and a weak stimulus (upper left corner in Figure 5). That is, the plots show that the effect of selectivity near zero stimulus-induced suppression is weak, although significant (main effect of selectivity at zero stimulus-induced suppression: cRF-cRF: p=2x10^-64^; sRF-cRF: p=2x10^-60^; M1: p=7x10^-136^ across RF configurations; M2: p=5x10^-30^ across RF configurations). "

*In addition, it is hard to reconcile Figure 5, where attention effects are strongly focused near selectivity = 1 (even if you collapse across suppression) with Figure 4, where attention effects are equally strong at selectivity = 0.5. What explains the difference?*

*Figure 4 shows the attention modulation as a function of selectivity, averaged across all data points with different suppression values. Note that there are more data points with a stimulus-induced suppression near zero (see Figure 4). For a stimulus-induced suppression near zero, attention modulation does not differ much across selectivity (see Figure 5). Because the average in Figure 4 is dominated by the majority of the data points, i.e. the data with stimulus-induced suppression near zero, attention modulation does not vary much as a function of selectivity above 0.5. Thus the average plots in Figure 4 hide important trends in the data, which only become apparent in Figure 5. The very strong attention effects indeed cluster near selectivity = 1 in Figure 5. This is because selectivity and stimulus-induced suppression interact with each other to determine attention modulation. To visualize this interaction, one needs to consider both variables (selectivity and suppression), as in Figure 5. The interaction causes attention modulation to increase rapidly with increasing values of selectivity and suppression, but fewer points exist near high selectivity and high suppression values (as expected because we could not optimize our stimuli for the majority of the simultaneously recorded neurons).*

*The authors respond that, in addition to the model, the claim is based on the visual similarity of Figure 5, and on "the statistical tests that we performed on these neuronal responses (see the discussion on the general linear model above)". 5A and B do not look entirely similar; in 5A the peak is elongated horizontally, showing that attention is more sensitive to suppression when the distractor is in the receptive field; in 5B the peak is elongated vertically, showing that attention is more sensitive to selectivity when the distractor is in the surround. A direct test of whether the attention effects are equivalent in 5A and 5B would be some kind of two-dimensional Kolmogorov-Smirnov analysis (see, e.g., Lopes, Reid, Hobson, Proceedings of Science, XI International Workshop on Advanced Computing and Analysis Techniques in Physics Research April 23-27 2007 Amsterdam, the Netherlands.) The linear regression is a less direct way of testing the question, and I don't think that it could capture the differences between peak shapes in 5A and 5B, in which case the new Bayes Factor analysis would not be meaningful.*

We thank the reviewer for providing us with this interesting article about two-dimensional Kolmogorov-Smirnov tests. This test compares two bivariate distributions. However, it is important to note that Figure 5 are not bivariate densities. Instead, Figure 5 plots attention modulation, not probability mass, as a function of selectivity and suppression. Thus to compare Figure 5 one would require a three-dimensional Kolmogorov-Smirnov test, with selectivity, suppression and attention modulation as variables. Crucially, this test will not give the correct answer to the question of similarity of attention effects between the receptive field and the surround conditions. This is because the test compares the full empirical distributions of both conditions, including whether the distributions of selectivity and stimulus-induced suppression indices differ between both receptive field configurations. We showed that the surround condition has on average higher selectivity values and lower suppression values than the classical receptive field condition (Figure 4). Thus, even if attention modulation follows the exact same principles in the classical receptive field and the surround, the test will detect differences due to different selectivities and suppressions between the two conditions.

*Importantly, Figure 5 shows that the spatially-tuned normalization model does capture most of the differences in peak shapes in Figure 5. This model has the same structure for the classical receptive field and the surround data, and only differs between these conditions by the numerical values of its parameters (see Figure 7). The single attention parameter (β) operates in exactly the same way in both receptive field configurations. So although the linear regression analysis is supportive of the claim, the strongest support comes from the very good fits of the attention modulations by the normalization model in both receptive-field configurations.*

*The authors explain that it is only the relative values between parameters that matter, and support this by writing out a multiplication of all terms by 1/alpha1. This still doesn't make sense to me in that the α values need to be able to go to 0, as in Figure 5/Figure 5—figure supplement 1, to represent absence of suppression. If suppression due to stimulus 1 is entirely absent (in which case the multiplier 1/alpha1 would be undefined), how would the model capture that? The addition of Figure 5/Figure 5—figure supplement 1 is certainly helpful in understanding the operation of the model, but it doesn't solve this confusion for me.*

Values of zero are well-approximated by very small values. In fact, the value of alpha2 in Figure 5/Figure 5—figure supplement 1 is not exactly zero, but 3x10^-16^. This is where the optimization algorithm stopped, because smaller values of alpha2 resulted in negligible improvements of the error function.

If suppression due the stimulus 1 is entirely absent, the algorithm increases the value of alpha2 so that the ratio alpha1/alpha2 becomes very small. So also here the relative values of the parameters matter. Specifically, the α of stimulus 1 is very small compared to that of stimulus 2.

*In their response, the authors provide an extended explanation of how α values can be balanced out in either direction by L values, concluding that "in general, there exists no direct relationship between the suppression-index in Figure 4 and the α term." To the extent this is true, Figure 7 and the accompanying legend are highly confusing, since they equate α with suppression in the labels and in the text. The fact that the new supplementary example with 0 suppression also has a 0 α value will tend to add to this confusion. Since they are plotting α in Figure 7, the authors need to explain what it means and how it relates (if at all) to suppression, not just to reviewers but also to readers.*

We thank the reviewer for pointing this out. We now write inthe Methods: "Note that the stimulus-induced suppression index is distinct from the αterms in the model. This is because the stimulus-induced suppression index is based on the observed neuronal responses, which comprise both an α and *L* term (i.e. response = *L/(*α+σ)). In terms of the model parameters, the stimulus-induced suppression index is given by: Stimulus−induced suppressionindex=responseP−responseP+NresponseP+responseP+N==LPαP+σ−LP+LNαP+αN+σLPαP+σ+LP+LNαP+αN+σ, where LPis the excitatory drive from the preferred component Gabor of a Gabor pair, LN is the excitatory drive from the non-preferred component Gabor, αP is the suppressive drive from the preferred component Gabor, and αN is the suppressive drive from the non-preferred component Gabor. So the stimulus-induced suppression index depends on both the excitatory and suppressive drive from the stimulus."

We also changed the legend of Figure 7 and write "Suppressive drive" and "Excitatory drive" instead of "Suppression" and "Excitation".

*The authors correctly point out that the models must not be failing to capture attention effects given the close approximation to population level effects in Figure 4. And, they reiterate that removing the attention term reduces overall explained variance from 87% to 79%. My main point had simply been that these were both extremely high values, and this is due to the fact that the number of terms in the model is on the same order as the number of conditions, so a close fit is not surprising or informative in either case. The cross-validation does not change this because, given the low number of conditions and the description now given in methods, it amounts to simply splitting repetitions from identical trial types into two groups and proving they give the same result. I don't think the 87% explained variance with the full model or the 8% drop in explained variance by themselves establish that "attention acts uniformly across the cRF and the surround". (The exceedingly close fits are due to the similar number of terms and conditions; the 8% drop shows that the attention term mattered, but it doesn't address how exactly attention behaved.) Nor, as explained above, do I think this is established by visual similarity between Figure 5 or the linear regression analysis. As stated above, I think the direct test of this proposition would be a statistical comparison of 5A and 5B. Regardless of the result, the clear and more interesting result here is the criticality of the interaction between selectivity and suppression for attention.*

As the reviewer correctly points out, a substantial amount of variance remains explained because of the variance that arises from stimulus differences. This is also because the responses of many neurons were little affected by attention (see neurons with weak selectivity and suppression in Figure 5), producing small decreases in explained variance when leaving out the β parameter. A smaller fraction of neurons were strongly affected by attention (see neurons with strong selectivity and suppression in Figure 5) and a model with no β parameter explained up to 50% of the variance *less* than a model with β parameter. If one were to select only highly selective neurons that are strongly suppressed by nearby stimuli, the average percentage of explained variance by the β parameter would be much higher. However, we could not optimize our stimuli for each of the simultaneously recorded neurons.

Our results show that the magnitude of attention modulation, and thus how much of the response variance is explained by attention and β, varies with suppression and selectivity. Thus the critical test of the model lies in its ability to explain the full range of attention modulations across the range of observed suppression and selectivity. We show that the model does an excellent job of fitting the observed attention modulations across the full range of selectivity and suppression values and in different stimulus and receptive field conditions. The model explains all these attention effects using a single attention parameter, i.e. β, that interacts with the spatial summation or normalization mechanisms of neurons. This β parameter can even be fixed across neurons and receptive field configurations, without appreciably affecting the average percentage explained variance. We believe this is a remarkable accomplishment of the model.

[Editors' note: further revisions were requested prior to acceptance, as described below.]

*Thank you for resubmitting your work entitled "Attention operates uniformly throughout the Classical Receptive Field and the Surround" for further consideration at eLife. Your revised article has been favorably evaluated by David Van Essen (Senior editor) and the Reviewing editor.*

*The manuscript has been improved and we greatly appreciate the detailed, equation-based clarifications in response to several of the points raised by the reviewers.*

*However, the reviewers remain concerned whether the authors have really provided evidence that attention acts uniformly throughout the receptive field. The reviewers would like clarification of the following points:*

*1) Is there any reason that 5A and 5B should look identical? In the initial reviewer response, the authors stated, "We would like to emphasize that the claim that attention acts in a uniform way on all cRF and surround positions is supported by more than just the model that accurately accounts for all neuronal responses, irrespective of the receptive field position and attention condition. In particular, this uniformity can be seen directly in the similarity of the neuronal responses in Figure 5, which show that the dependency of attention modulation on selectivity and suppression does not depend on whether stimuli are presented in the cRF or the surround." The reviewers see a **clear difference between 5A and B**. This seems to provide evidence that attention is more critically dependent on selectivity in the CRF-CRF condition, and more critically dependent on suppression in the CRF-SRF condition; quite the opposite of operating uniformly. The reviewers are confused whether the authors have a hypothesis or explanation for why the dependency of attention on suppression and selectivity should differ between CRF (5A) and SRF (5B); it seems that the difference is contrary to the authors' conclusion. If they do have a hypothesis about the difference, please state it and test it, and modify the "uniform operation" conclusion accordingly.*

In pointing out the striking similarity between Figure 5, we did not mean to claim that receptive field and surround mechanisms were identical in every detail, or that CRF and surround phenomena depend on exactly the same circuitry (which seems unlikely), or that there is pixel-by-pixel identity between the two plots. We meant to emphasize that in both cases attention-related modulation depends critically on a combination of selectivity and suppression in a quantitatively similar way in both the CRF and surround. We stand by that claim, and emphasize that this powerful first-order similarity has never before been reported in the literature.

It is conceivable that there are some quantitative differences in the roles of selectivity and suppression between the receptive field and the surround, such as those described by the reviewers. Alternatively, the small second-order differences in Figure 5 might depend, at least partially, on sampling noise, as suggested by the statistical tests (see below). We will continue to work on refining our models as we collect data in the future, but differences of that sort will not change the first-order observations we describe here and are well beyond the scope of the current report.

We have modified the text to make the specific claims clearer (sixth paragraph of Discussion): "Suppression and excitation may rely on distinct mechanisms in different regions of the receptive field^31^. Our data do not pertain to these different mechanisms and we may have missed some small differences in attention modulation associated with these distinct mechanisms. Nonetheless, our findings show that the way attention interacts with excitation and suppression across different regions of the receptive field is remarkably similar."

*2) In the most recent reviewer response, the authors seem to be backing away from claim that 5A and B need to look similar, and are appealing to the similarity between the model fits (5D,E) and the data (5A, B). The reviewers remain concerned that the reason why the model fits assuming uniform attention are so good is that all of the non-attention related terms (separate excitatory term for each stimulus type/location combination and a separate suppression value for each location) are doing all the work. Since the number of terms being fitted is nearly equal to the number of measurements, this guarantees over fitting, and it can't be cured by cross-validation because every condition would need to be in both portions of the data to fit all the stimulus- and position-specific terms.*

*It would seem advisable to include an explicit caveat about this.*

Please note that the non-attention related terms are not doing all the work in the model. Without the attention term (β), the model predicts no attention modulation. Consequently, without the attention term Figure 5 would be uniformly zero, in clear contradiction to the data. Hence the attention term is critical in explaining all attention modulations. Thus in addition to the non-attention terms, the attention term performs crucial work.

Please also note that our model has nine free parameters (constant β) that account for 36 measurements in each neuron (see Methods). Thus the number of terms (9) is not nearly equal to the number of measurements (36).

Finally, we point out that cross-validation does in fact counteract the adverse effects of over fitting. Cross validation gives a nearly unbiased estimate of a model's performance on the validating data. In particular, a model with too many terms is expected to perform worse on cross-validation because the fits on one data half would include fitted noise (i.e. over fitting) and this fitted noise is independent of the noise in the validating data, causing lower explained variances. Thus with too many model terms the cross validation would punish, not improve, the goodness-of-fit measures.

*3) Most importantly, the obvious way to test their claim is some kind of direct comparison of attentional modulation between CRF and SRF. Can the authors do this?*

We provide several such direct comparisons between attention modulations in the CRF and the surround.

First, there is no significant interaction between RF configuration (CRF and SRF) and the effects of selectivity and suppression on attention modulation (p ≥ 0.6; Results section, subsection “Stimulus selectivity and stimulus-induced suppression interact in determining attention modulation and do so similarly inside the cRF and the surround”). In other words, attention modulation as a function of selectivity and suppression does not differ significantly between the cRF and the surround. Note that this is a direct test of the hypothesis that attention modulation as a function of selectivity and suppression is the same in the cRF and surround, without appeal to goodness of model fits. We emphasize that classical statistical tests (i.e. direct tests) are not designed to test the veracity of a null-hypothesis such as, in our case, identical attention modulation in the CRF and the surround: a non-significant p-value provides little information about the truthfulness of the null hypothesis. This is why we also included a Bayesian test, which does not suffer from this drawback.

Second, the Bayesian analysis showed that the data are much (347 times) more likely to come from a model in which one does not distinguish between RF configurations. According to this Bayesian analysis, adding extra parameters to distinguish between the CRF and SRF would be spurious and cause over fitting. Thus this Bayesian analysis indicates that attention modulation as a function of selectivity and suppression is the same in the cRF and surround.

Third, the spatially-tuned normalization model, which has no extra parameters to distinguish between RF configurations, successfully accounts for all data in the CRF and the surround.

In empirical sciences, such as neuroscience, one cannot prove statements, one can only provide evidence in favor of a hypothesis. The three above-mentioned analyses all point in the same direction, namely that it is most probable and parsimonious to conclude that attention modulation operates uniformly across the CRF and SRF.

*4) Have the authors tried to fit other models with fewer non-attention related variables (e.g., keeping suppression constant)? How does this affect the attention term?*

*Overall, the reviewers would be most convinced by a new analysis directly showing that attentional modulation as a function of selectivity and suppression is the same in the cRF and surround, without appeal to goodness of model fits. Without such an analysis, it remains unclear whether the authors have identified a truth about the brain, that attention acts uniformly, or whether they have simply shown that a viable model of neurons can be constructed in which attention acts uniformly, but other models with a variable attention term are equally plausible.*

We have explored other models. Two of these analyses were included in the previous versions of the manuscript. We have also added in the current version an additional analysis in which we fitted a model with constant excitatory terms.

Results section, subsection “Spatial variability in excitation and suppression underlies differences in attention modulation across neurons” in the manuscript we describe the model fits with constant suppression terms. We repeat the conclusion here: a model with constant suppression terms, but with free attention term, fits the data significantly worse. This analysis, together with the spatially-tuned suppression shown in Figure 7 and findings from previous studies, demonstrates that a separate suppression term for each spatial location was necessary to account for the observed neuronal responses.

On the “Model” subsection of the Materials and methods section, we describe the model fits with no σ term. We repeat the conclusion here: a model with no σ term fits the data significantly worse. So we included a σ term in the model.

We have added the results from a new analysis in which we fit for each neuron a model with only one free excitatory (*L*) term to capture excitation across all stimulus conditions. This model with one *L* term performed significantly worse at explaining neuronal responses (median two-fold cross-validated percentage explained variance 48%, compared to 87% for the model with all *L* terms; p < 0.0001; sequential F-test).

Taken together, these analyses show that all non-attention terms were necessary to account for the observed neuronal responses.

When fitting models in which some of these non-attention terms are omitted, the attention term may stay unchanged in some cases, i.e. for neurons in which the omitted non-attention term happened to be relatively unimportant given the stimulus conditions, or change in other (most) cases, i.e. trying to compensate for the detrimental effects of omitting necessary non-attention terms. In any case, the changes in the attention term depend on a complex interaction between the specific neuron, stimulus conditions and the type of the omitted parameter. Importantly, these changes in the attention term are meaningless because the attention terms now tries to capture some of the effects of an omitted non-attention term, so it loses its relationship to attention per se.